# The Pursuit of Human Labeling:
# A New Perspective on Unsupervised Learning

**Artyom Gadetsky**
EPFL
artem.gadetskii@epfl.ch

**Maria Brbić**
EPFL
mbrbic@epfl.ch

## Abstract

We present HUME, a simple model-agnostic framework for inferring human labeling of a given dataset without any external supervision. The key insight behind our approach is that classes defined by many human labelings are linearly separable regardless of the representation space used to represent a dataset. HUME utilizes this insight to guide the search over all possible labelings of a dataset to discover an underlying human labeling. We show that the proposed optimization objective is strikingly well-correlated with the ground truth labeling of the dataset. In effect, we only train linear classifiers on top of pretrained representations that remain fixed during training, making our framework compatible with any large pretrained and self-supervised model. Despite its simplicity, HUME outperforms a supervised linear classifier on top of self-supervised representations on the STL-10 dataset by a large margin and achieves comparable performance on the CIFAR-10 dataset. Compared to the existing unsupervised baselines, HUME achieves state-of-the-art performance on four benchmark image classification datasets including the large-scale ImageNet-1000 dataset. Altogether, our work provides a fundamentally new view to tackle unsupervised learning by searching for consistent labelings between different representation spaces.

## 1 Introduction

A key aspect of human intelligence is an ability to acquire knowledge and skills without external guidance or instruction. While recent self-supervised learning methods [1, 2, 3, 4, 5, 6] have shown remarkable ability to learn task-agnostic representations without any supervision, a common strategy is to add a linear classification layer on top of these pretrained representations to solve a task of interest. In such a scenario, neural networks achieve high performance on many downstream human labeled tasks. Such strategy has also been widely adopted in transfer learning [7, 8] and few-shot learning [9, 10], demonstrating that a strong feature extractor can effectively generalize to a new task with a minimal supervision. However, a fundamental missing piece in reaching human-level intelligence is that machines lack an ability to solve a new task without any external supervision and guidance.

Close to such ability are recent multi-modal methods [6, 11, 12] trained on aligned text-image corpora that show outstanding performance in the zero-shot learning setting without the need for fine-tuning. However, zero-shot learning methods still require human instruction set to solve a new task. In a fully unsupervised scenario, labels for a new task have been traditionally inferred by utilizing clustering methods [13, 14, 15, 16], designed to automatically identify and group samples that are semantically related. Compared to (weakly) supervised counterparts, the performance of clustering methods is still lagging behind.

In this work, we propose HUME, a simple model-agnostic framework for inferring human labeling of a given dataset without any supervision. The key insights underlying our ap-

37th Conference on Neural Information Processing Systems (NeurIPS 2023).

proach are that: *(i)* many human labeled tasks are *linearly separable* in a sufficiently strong representation space, and *(ii)* although deep neural networks can have their own inductive biases that do not necessarily reflect human perception and are vulnerable to fitting spurious features [17, 18], human labeled tasks are *invariant* to the underlying model and resulting representation space. We utilize these observations to develop the generalization-based optimization objective which is strikingly well correlated with human labeling (Figure 1).

The key idea behind this objective is to evaluate the generalization ability of linear models on top of representations generated from two pretrained models to assess the quality of any given labeling (Figure 2). Our framework is model-agnostic, *i.e.*, compatible with any pretrained representations, and simple, *i.e.*, it requires training only linear models.

Overall, HUME presents a new look on how to tackle unsupervised learning. In contrast to clustering methods [13, 14] which try to embed inductive biases reflecting semantic relatedness of samples into a learning algorithm, our approach addresses this setting from model generalization perspective. We instantiate HUME's framework using representations from different self-supervised methods (MOCO [4, 3], SimCLR [1, 2], BYOL [19]) pretrained on the target dataset and representations obtained using large pretrained models (BiT [7], DINO [20], CLIP [6]). Remarkably, despite being fully unsupervised, HUME outperforms a supervised linear classifier on the STL-10 dataset by $5\%$ and has comparable performance to a linear classifier on the CIFAR-10 dataset. Additionally, it leads to the new state-of-the-art performance on standard clustering benchmarks including the CIFAR-100-20 and the large-scale ImageNet-1000 datasets. Finally, our framework can construct a set of reliable labeled samples transforming the initial unsupervised learning problem to a semi-supervised learning.

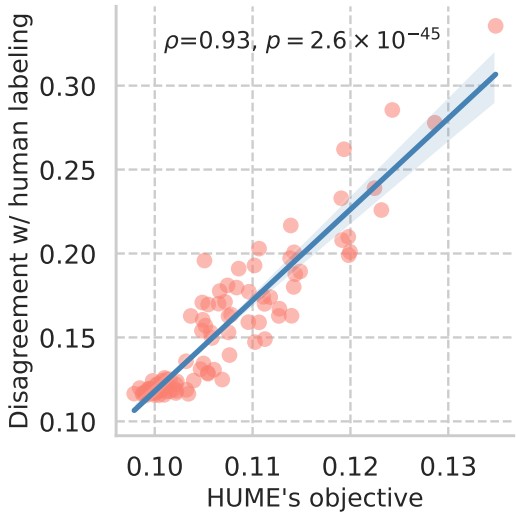

Figure 1: Correlation plot between distance to ground truth human labeling and HUME's objective on the CIFAR-10 dataset. HUME generates different labelings of the data to discover underlying human labeling. For each labeling (data point on the plot), HUME evaluates generalization error of linear classifiers in different representation spaces as its objective function. HUME 's objective is strikingly well correlated ($\rho = 0.93, p = 2.6 \times 10^{-45}$ two-sided Pearson correlation coefficient) with a distance to human labeling. In particular, HUME achieves the lowest generalization error for tasks that almost perfectly correspond to human labeling, allowing HUME to recover human labeling without external supervision. Results on the STL-10 and CIFAR-100-20 datasets are provided in Appendix D.

## 2 HUME framework

In this section, we first introduce our problem setting and then present a general form of our framework for finding human labeled tasks without any supervision.

**Problem setting**. Let $\mathcal{D} = \{x_i\}_{i=1}^N$ be a set of samples. We assume this dataset consists of $K$ classes, where $K$ is known a priori, and each example $x_i$ can belong only to one particular class $k \in \{0, \ldots, K-1\}$. We define a task $\tau : \mathcal{D} \to \{0, \ldots, K-1\}$ as a labeling function of this dataset. We refer to a task $\tau$ as human labeled if it respects the true underlying labeling of the corresponding dataset $\mathcal{D}$.

### 2.1 Test error and invariance of human labeled tasks to representation space

Measuring the performance of a model on a held-out dataset is a conventional method to assess an ability of the model to generalize on the given task $\tau$. Specifically, for the dataset $\mathcal{D}$, we can construct two disjoint subsets $(X_{tr}, X_{te})$ of the dataset $\mathcal{D}$. Let $f : \mathcal{D} \to \Delta^{K-1}$ be a probabilistic classifier which transforms the input $x \in \mathcal{D}$ to class probabilities, *i.e.*, $\Delta^{K-1}$ is a $(K-1)$-dimensional simplex. After training $f$ on $X_{tr}$ with loss function $\mathcal{L}$ and labeling $\tau(X_{tr})$, we can compute the test error on

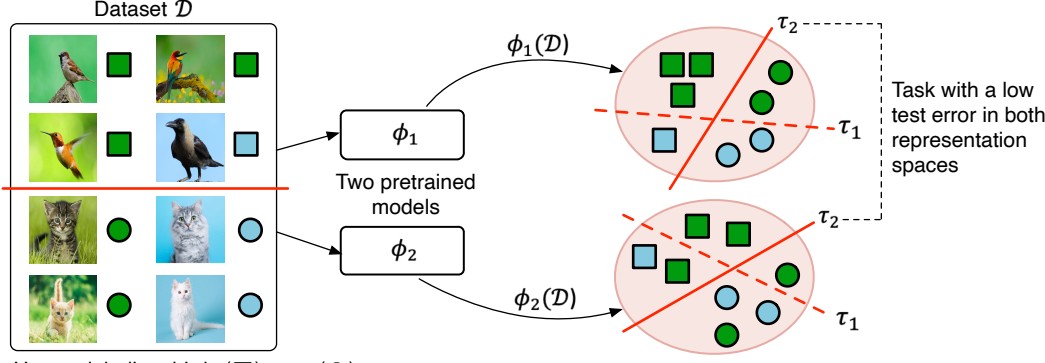

Figure 2: Overview of the HUME framework. HUME utilizes pretrained representations and linear models on top of these representations to assess the quality of any given labeling. As a result, optimizing the proposed generalization-based objective leads to labelings which are strikingly well correlated with human labelings.

$X_{te}$ which can provide us with an unbiased estimate of the true error of the model on the task $\tau$:

$$\mathcal{L}(f(X_{te}), \tau(X_{te})) = \frac{1}{|X_{te}|} \sum_{x \in X_{te}} \mathcal{L}(f(x), \tau(x)). \tag{1}$$

In HUME, we utilize this score to measure the quality of any given task $\tau$. By utilizing this score, we aim at searching for a human labeled task over the set of all possible tasks on the dataset $\mathcal{D}$. However, the main challenge is that neural networks can have their own inductive biases and attain low test error on tasks that capture spurious correlations and do not reflect human labeling [17]. To distinguish between such tasks and human labeled tasks, the key insight behind our framework is that for many human labeled tasks classes defined by human labeling are linearly separable regardless of the representation space used to represent a dataset. In other words, human-labeled tasks are *invariant* to *sufficiently strong* representation spaces. We next formally define what we mean by a sufficiently strong representation space and invariance of a task to the pair of representations.

**Definition 1.** *Let $\phi(x) : \mathcal{D} \to \mathbb{R}^d$ be a mapping from the sample space to a low-dimensional representation space. We say that a representation space is sufficiently strong with respect to $\tau$ if a linear model $f$ trained on top of $\phi(\cdot)$ attains low test error in Eq. (1).*

**Definition 2.** *Let $\phi_1(x) : \mathcal{D} \to \mathbb{R}^{d_1}$ and $\phi_2(x) : \mathcal{D} \to \mathbb{R}^{d_2}$ be two mappings from the sample space to low-dimensional representations. We say that a task $\tau$ is invariant to the pair of representations $(\phi_1(x), \phi_2(x))$ if both representation spaces are linearly separable with respect to $\tau$, i.e., both linear models $f_1$ and $f_2$, trained on top of $\phi_1(\cdot)$ and $\phi_2(\cdot)$ respectively, attain low test error in Eq. (1).*

We employ the property of invariance to the given pair of *fixed pretrained representations* to seek for a human labeled task. Thus, we only train linear classifiers on top pretrained representations while the representations are always frozen during training. The simultaneous utilization of several representation spaces acts as a regularizer and prevents learning tasks that capture spurious correlations which can reflect inductive biases of the individual representation space.

Specifically, given a task $\tau$, we aim to fit a linear model $f_i$ with weights parametrized by $W_i \in \mathbb{R}^{K \times d_i}$ on $X_{tr}$ in each representation space $\phi_i(\cdot)$, $i = 1, 2$. Let $\hat{W}_i(\tau)$ be the solution for the corresponding $\phi_i(\cdot)$. We aim at optimizing the test error of both linear models with respect to $\tau$:

$$\arg \min_{\tau} \mathcal{L}(\sigma(\hat{W}_1(\tau)\phi_1(X_{te})), \tau(X_{te})) + \mathcal{L}(\sigma(\hat{W}_2(\tau)\phi_2(X_{te})), \tau(X_{te})), \tag{2}$$

where $\sigma(\cdot)$ is a softmax activation function. We draw an attention of the reader to the fact that both $\hat{W}_1(\tau)$ and $\hat{W}_2(\tau)$ implicitly depend on $\tau$ as solutions of the *inner* optimization problem on $X_{tr}$ with labeling $\tau(X_{tr})$. We discuss how to efficiently solve this optimization problem and propagate gradients through the optimization process in the corresponding Section 2.3.

An unresolved modeling question is the choice of the representation spaces $\phi_{1,2}(\cdot)$. We utilize self-supervised pretraining on the target dataset $\mathcal{D}$ to obtain robust and well clustered representations

for representation space $\phi_1(\cdot)$. Representation space $\phi_2(\cdot)$ acts as a regularizer to guide the search process. Thus, we utilize features of a large pretrained model as the representation space $\phi_2(\cdot)$. This is an appealing modeling design choice from both efficiency and model performance perspective. In particular, by using a large pretrained model we do not need to train a model on the given dataset of interest. Despite the simplicity, the linear layer fine-tuning on top of the fixed representations of the deep pretrained models has shown its efficiency in solving many downstream problems [6, 20, 7, 10]. The approach to model $\tau$ is discussed in the next section.

## 2.2 Task parametrization

The proposed objective in Eq. (2) requires solving difficult discrete optimization problem with respect to $\tau$ which prevents us from using efficient gradient optimization techniques. Additionally, it requires designing three separate models $(\phi_1(\cdot), \phi_2(\cdot), \tau(\cdot))$ which can be computationally expensive and memory-intensive in practice. To alleviate both shortcomings, we utilize $\phi_1(\cdot)$ to simultaneously serve as a basis for the task encoder and as a space to which the task should be invariant to. Namely, we relax the outputs of $\tau$ to predict class probabilities instead of discrete class assignments, and parametrize the task $\tau_{W_1}(\cdot) : \mathcal{D} \to \Delta^{K-1}$ as follows:

$$\tau_{W_1}(x) = \mathcal{A}(W_1\hat{\phi}_1(x)), \; W_1W_1^T = I_K, \; \hat{\phi}_1(x) = \frac{\phi_1(x)}{\|\phi_1(x)\|_2}, \tag{3}$$

where $\phi_1(\cdot)$ denotes the self-supervised representations pretrained on the given dataset $\mathcal{D}$ and these representations also remain *fixed* during the overall training procedure. We produce sparse labelings using sparsemax [21] activation function $\mathcal{A}$ since each sample $x_i \in \mathcal{D}$ needs to be restricted to belong to a particular class. The above parametrization may be viewed as learning prototypes for each class which is an attractive modeling choice for the representation space $\hat{\phi}_1(\cdot)$ [22, 9]. Thus, $W_1\hat{\phi}_1(\cdot)$ corresponds to the cosine similarities between class prototypes $W_1$ and the encoding of the sample $\hat{\phi}_1(\cdot)$. Moreover, sparsemax activation function acts as a soft selection procedure of the closest class prototype. Eventually, it can be easily seen that any linear dependence $W_1\hat{\phi}_1(x)$ give rise to the task which, by definition, is at least invariant to the corresponding representation space $\hat{\phi}_1(x)$.

Given the above specifications, our optimization objective in Eq. (2) is simplified as follows:

$$\arg\min_{W_1} \mathcal{L}(\sigma(\hat{W}_2(W_1)\phi_2(X_{te})), \tau_{W_1}(X_{te})), \tag{4}$$

where $\hat{W}_2(W_1)$ denotes the weights of the linear model $f_2$ trained on the $(X_{tr}, \tau_{W_1}(X_{tr}))$, which implicitly depend on the parameters $W_1$. We use the cross-entropy loss function $\mathcal{L}$, which is a widely used loss function for classification problems. The resulting optimization problem is continuous with respect to $W_1$, which allows us to leverage efficient gradient optimization techniques. Although it involves propagating gradients through the inner optimization process, we discuss how to efficiently solve it in the subsequent section.

## 2.3 Test error optimization

At each iteration $k$, we randomly sample disjoint subsets $(X_{tr}, X_{te}) \sim D$ to prevent overfitting to the particular train-test split. We label these splits using the current task $\tau_{W_1^k}(\cdot)$ with parameters $W_1^k$. Before computing the test risk defined in Eq. (4), we need to solve the inner optimization problem on $(X_{tr}, \tau_{W_1^k}(X_{tr}))$, specifically:

$$\hat{W}_2(W_1^k) = \arg\min_{W_2} \mathcal{L}(\sigma(W_2\phi_2(X_{tr})), \tau_{W_1^k}(X_{tr})). \tag{5}$$

It can be easily seen that the above optimization problem is the well-studied multiclass logistic regression, which is convex with respect to $W_2$ and easy to solve. To update parameters $W_1^k$, we need to compute the total derivative of Eq. (4) with respect to $W_1$ which includes the Jacobian $\frac{\partial \hat{W}_2}{\partial W_1^k}$. Different approaches [23, 24, 25, 26] can be utilized to compute the required Jacobian and propagate gradients through the above optimization process. For simplicity, we run gradient descent for the fixed number of iterations $m$ to solve the inner optimization problem and obtain $\hat{W}_2^m(W_1^k)$. Afterwards, we employ MAML [27] to compute $\frac{\partial \hat{W}_2^m}{\partial W_1^k}$ by unrolling the computation graph of the

inner optimization's gradient updates. The remaining terms of the total derivative are available out-of-the-box using preferred automatic differentiation (AD) toolbox. This results in the efficient procedure which can be effortlessly implemented in existing AD frameworks [28, 29].

**Regularization.** The task encoder $\tau$ can synthesize degenerate tasks, *i.e.*, assign all samples to a single class. Although such tasks are invariant to any representation space, they are irrelevant. To avoid such trivial solutions, we utilize entropy regularization to regularize the outputs of the task encoder averaged over the set $X = X_{tr} \cup X_{te}$, specifically

$$\mathcal{R}(\overline{\tau}) = - \sum_{k=1}^{K} \overline{\tau}_k \log \overline{\tau}_k, \tag{6}$$

where $\overline{\tau} = \frac{1}{|X|} \sum_{x \in X} \tau_\theta(x) \in \Delta^{K-1}$ is the empirical label distribution of the task $\tau$. This leads us to the final optimization objective, which is:

$$\arg \min_{W_1} \mathcal{L}(\sigma(\hat{W}_2(W_1)\phi_2(X_{te})), \tau_{W_1}(X_{te})) - \eta \mathcal{R}(\overline{\tau}(W_1)), \tag{7}$$

where $\eta$ is the regularization parameter. This regularization corresponds to entropy regularization which has been widely used in previous works [13, 30, 31]. The pseudocode of the algorithm is shown in Algorithm 1.

---

**Algorithm 1** HUME: A simple framework for finding human labeled tasks

---

**Input:** Dataset $\mathcal{D}$, number of classes $K$, number of iterations $T$, representation spaces $\phi_{1,2}(\cdot)$, task encoder $\tau_{W_1}(\cdot)$, regularization parameter $\eta$, step size $\alpha$
1: Randomly initialize $K$ orthonormal prototypes: $W_1^1 = \text{ortho\_rand(K)}$
2: **for** $k = 1$ to $T$ **do**
3:     Sample disjoint train and test splits: $(X_{tr}, X_{te}) \sim \mathcal{D}$
4:     Generate task: $\tau_{tr}, \tau_{te} \leftarrow \tau_{W_1^k}(X_{tr}), \tau_{W_1^k}(X_{te})$
5:     Fit linear classifier on $X_{tr}$: $\hat{W}_2(W_1^k) = \arg \min_{W_2} \mathcal{L}(\sigma(W_2\phi_2(X_{tr})), \tau_{tr})$
6:     Evaluate task invariance on $X_{te}$: $\mathcal{L}_k(W_1^k) = \mathcal{L}(\sigma(\hat{W}_2(W_1^k)\phi_2(X_{te})), \tau_{te}) - \eta \mathcal{R}(\overline{\tau})$
7:     Update task parameters: $W_1^{k+1} \leftarrow W_1^k - \alpha \nabla_{W_1^k} \mathcal{L}_k(W_1^k)$
8: **end for**

---

## 3 Experiments

### 3.1 Experimental setup

**Datasets and evaluation metrics.** We evaluate the performance of HUME on three commonly used clustering benchmarks, including the STL-10 [32], CIFAR-10 and CIFAR-100-20 [33] datasets. The CIFAR-100-20 dataset consists of superclasses of the original CIFAR-100 classes. In addition, we also compare HUME to large-scale unsupervised baselines on the fine-grained ImageNet-1000 dataset [34]. We compare our method with the baselines using two conventional metrics, namely clustering accuracy (ACC) and adjusted rand index (ARI). Hungarian algorithm [35] is used to match the found labeling to the ground truth labeling for computing clustering accuracy. We interchangeably use terms generalization error and cross validation accuracy when presenting the results.

**Instantiation of HUME.** For the representation space $\phi_1(\cdot)$, we use MOCOv2 [4] pretrained on the train split of the corresponding dataset. We experiment with the SimCLR [1] as a self-supervised method in Appendix C. For the representation space $\phi_2(\cdot)$, we consider three different large pretrained models: *(i)* BiT-M-R50x1 [7] pretrained on ImageNet-21k [36], *(ii)* CLIP ViT-L/14 pretrained on WebImageText [6], and *(iii)* DINOv2 ViT-g/14 pretrained on LVD-142M [20].

**Baselines.** Since HUME trains linear classifiers on top of pretrained self-supervised representations, supervised linear probe on top of the same self-supervised pretrained representations is a natural baseline to evaluate how well the proposed framework can match the performance of a supervised model. Thus, we train a linear model using ground truth labelings on the train split and report the

results on the test split of the corresponding dataset. For the unsupervised evaluation, we consider two state-of-the-art deep clustering methods, namely SCAN [13] and SPICE [14]. Both methods can be seen as three stage methods. First stage employs self-supervised methods to obtain good representations. We consistently use the ResNet-18 backbone pretrained with MOCOv2 [4] for all baselines as well as HUME. During the second stage these methods perform clustering on top of the frozen representations and produce reliable pseudo-labels for the third stage. Finally, the third stage involves updating the entire network using generated pseudo-labels. Thus, pseudo-labels are produced using a clustering algorithm from the second step and third stage is compatible with applying any semi-supervised method (SSL) [37, 38] on the set of reliable samples. Instead of optimizing for performance of different SSL methods which is out-of-scope of this work, we compare clustering performance of different methods and report the accuracy of generated pseudo-labels which is a crucial component that enables SSL methods to be effectively applied. Additionally, we utilize the recent state-of-the-art SSL method FreeMatch [39] to study the performance of FreeMatch when applied to HUME's reliable samples. As additional unsupervised baselines, we include results of K-means clustering [15] on top of the corresponding representations and K-means clustering on top of concatenated embeddings from both representation spaces used by HUME. For stability, all K-means results are averaged over 100 runs for each experiment. On the ImageNet-1000 dataset, we compare HUME to the recent state-of-the-art deep clustering methods on this benchmark. Namely, in addition to SCAN, we also consider two single-stage methods, TWIST [40] and Self-classifier [41]. TWIST is trained from scratch by enforcing consistency between the class distributions produced by a siamese network given two augmented views of an image. Self-classifier is trained in the similar fashion as TWIST, but differs in a way of avoiding degenerate solutions, *i.e.*, assigning all samples to a single class. HUME can be trained in inductive and transductive settings: *inductive* corresponds to training on the train split and evaluating on the held-out test split, while *transductive* corresponds to training on both train and test splits. Note that even in transductive setting evaluation is performed on the test split of the corresponding dataset to be comparable to the performance in the inductive setting. We report transductive and inductive performance of HUME when comparing it to the performance of the supervised classifier. For consistency with the prior work [13], we evaluate clustering baselines in the inductive setting.

**Implementation details.** For each experiment we independently run HUME with 100 different random seeds and obtain 100 different labelings. To compute the labeling agreement for the evaluation, we simply use the Hungarian algorithm to match all found labelings to the labeling with the highest cross-validation accuracy (HUME's objective). Finally, we aggregate obtained labelings using majority vote, *i.e.*, the sample has class $i$ if the majority of the found labelings predicts class $i$. We show experiments with different aggregation strategies in Section 3.2. We set the regularization parameter $\eta$ to 10 in all experiments. We show robustness to this hyperparameter in Appendix B. We provide other implementation details in Appendix A. Code is publicly available at `https://github.com/mlbio-epfl/hume`.

## 3.2  Results

**Comparison to supervised baseline.** We compare HUME to a supervised linear classifier by utilizing ResNet-18 MOCOv2 pretrained representations [4] for both models. In particular, for a supervised classifier we add a linear layer on top of pretrained representations which is standard evaluation strategy of self-supervised methods. As a regularization representation space in HUME, we utilize BiT [7], CLIP ViT [6] and DINOv2 ViT [20]. The results are shown in Table 1. Remarkably, without using any supervision, on the STL-10 dataset HUME consistently achieves better performance than the supervised linear model in the transductive setting, and using CLIP and DINO in the inductive setting. Specifically, using the strongest DINO model, HUME outperforms the supervised linear model by $5\%$ on the STL-10 dataset in terms of accuracy and by $11\%$ in terms of ARI. On the CIFAR-10 dataset, HUME achieves performance comparable to the linear classifier. On the hardest CIFAR-100-20 there is still an expected gap between the performance of supervised and unsupervised methods. When comparing performance of HUME in inductive and transductive setting, the results show that utilizing more data in the transductive setting consistently outperforms the corresponding method in inductive setting by $1 - 3\%$ in terms of accuracy. Finally, when comparing performance of different pretrained models, the results show that employing larger pretrained models results in better performance. For example, on the CIFAR-100-20 dataset HUME DINO achieves $16\%$ relative improvement in accuracy over HUME BiT in both inductive and transductive settings. Thus,

these results strongly indicate that HUME framework can achieve even better performance by taking advantage of unceasing progress in the development of large pretrained models.

Table 1: Comparison of HUME to a supervised linear classifier in inductive (ind) and transductive (trans) settings using MOCOv2 self-supervised representations pretrained on the corresponding dataset and three different large pretrained models.

| Method | STL-10 | | CIFAR-10 | | CIFAR-100-20 | |
|---|---|---|---|---|---|---|
| | ACC | ARI | ACC | ARI | ACC | ARI |
| **MOCO Linear** | 88.9 | 77.7 | **89.5** | 79.0 | **72.5** | **52.6** |
| **HUME BiT ind** | 87.5 | 76.2 | 85.9 | 73.8 | 47.8 | 33.5 |
| **HUME CLIP ind** | 90.2 | 80.2 | 88.2 | 77.1 | 48.5 | 34.1 |
| **HUME DINO ind** | 90.8 | 81.2 | 88.4 | 77.6 | 55.5 | 37.7 |
| **HUME BiT trans** | 90.3 | 80.5 | 86.6 | 75.0 | 48.8 | 34.5 |
| **HUME CLIP trans** | 92.2 | 84.1 | 88.9 | 78.3 | 50.1 | 34.8 |
| **HUME DINO trans** | **93.2** | **86.0** | 89.2 | 79.2 | 56.7 | 39.6 |

**Comparison to unsupervised baselines.** We next compare performance of HUME to the state-of-the-art deep clustering methods (Table 2). We use DINOv2 as the second representation space but the results for other models are available in Table 1. Results show that HUME consistently outperforms all baseline by a large margin. On the STL-10 and CIFAR-10 datasets, HUME achieves 5% improvement in accuracy over the best deep clustering baseline and 11% and 10% in ARI, respectively. On the CIFAR-100-20 dataset, HUME achieves remarkable improvement of 19% in accuracy and 18% in ARI. It is worth noting that considered baselines utilize nonlinear models (multilayer perceptrons) on top of the pretrained representations, while HUME employs solely a linear model. When compared to the K-means clustering baselines on top of concatenated features from DINO and MOCO, HUME achieves 18%, 9% and 8% relative improvement in accuracy on the STL-10, CIFAR-10 and CIFAR-100-20 datasets, respectively. These results demonstrate that performance gains come from HUME's objective rather than from utilizing stronger representation spaces. Overall, our results show that the proposed framework effectively addresses the challenges of unsupervised learning and outperforms other baselines by a large margin.

Table 2: Comparison to unsupervised baselines. All methods use ResNet-18 MOCOv2 self-supervised representations pretrained on the target dataset. We use DINOv2 as a large pretrained model.

| Method | STL-10 | | CIFAR-10 | | CIFAR-100-20 | |
|---|---|---|---|---|---|---|
| | ACC | ARI | ACC | ARI | ACC | ARI |
| **MOCO + K-means** | 67.7 | 54.1 | 66.9 | 51.8 | 37.5 | 20.9 |
| **DINO + K-means** | 60.1 | 35.4 | 75.5 | 67.6 | 47.2 | 33.9 |
| **DINO + MOCO + K-means** | 77.1 | 70.0 | 81.4 | 77.1 | 51.3 | 36.1 |
| **SCAN** | 77.8 | 61.3 | 83.3 | 70.5 | 45.4 | 29.7 |
| **SPICE** | 86.2 | 73.2 | 84.5 | 70.9 | 46.8 | 32.1 |
| **HUME** | **90.8** | **81.2** | **88.4** | **77.6** | **55.5** | **37.7** |

**Large-scale ImageNet-1000 benchmark.** We next study HUME's performance on the ImageNet-1000 benchmark and compare it to the state-of-the-art deep clustering methods on this large-scale benchmark. All methods use the ResNet-50 backbone. SCAN is trained using MOCOv2 self-supervised representations and both TWIST and Self-classifier are trained from scratch as single-stage methods. HUME utilizes the same MOCOv2 self-supervised representations and DINOv2 as the second representation space. The results in Table 3 show that HUME achieves 24% relative improvement in accuracy and 27% relative improvement in ARI over considered baselines, thus confirming the scalability of HUME to challenging fine-grained benchmarks.

Table 3: Performance of HUME on the large-scale ImageNet-1000 dataset and comparison to unsupervised baselines. All methods use the ResNet50 backbone. HUME uses DINOv2 large pretrained model and ResNet-50 MOCOv2 self-supervised representation.

| Method | ACC | ARI |
|---|---|---|
| **SCAN** | 39.7 | 27.9 |
| **TWIST** | 40.6 | 30.0 |
| **Self-classifier** | 41.1 | 29.5 |
| **HUME** | **51.1** | **38.1** |

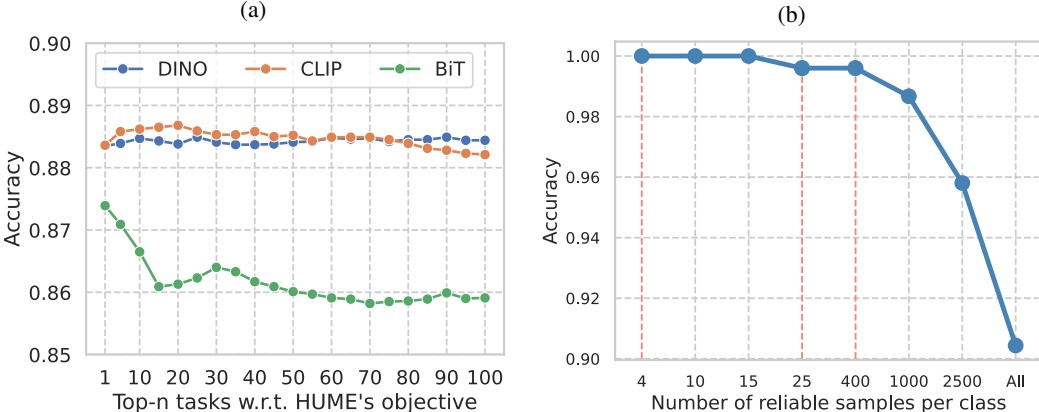

Figure 3: **(a)** Different aggregation strategies on the CIFAR-10 dataset. We use MOCOv2 and different large pretrained models to instantiate HUME. **(b)** Accuracy of the reliable samples on the CIFAR-10 dataset. We use MOCOv2 and DINOv2 to instantiate HUME. The well-established setting for testing SSL methods on the CIFAR-10 dataset uses 4, 25 and 400 reliable samples per class (depicted with lines in red color).

**Ablation study on aggregation strategy.** HUME achieves a strikingly high correlation between its objective function and ground truth labeling (Figure 1). However, due to the internal stochasticity, the found labeling with the lowest generalization error does not need to always correspond to the highest accuracy. Thus, to stabilize the predictions we obtain final labeling by aggregating found labelings from independent runs. We simply use the majority vote of all tasks in all our experiments. Here, we investigate the effect of using different aggregation strategies on the performance. Figure 3 (a) shows the results on the CIFAR-10 dataset and the corresponding plots for the STL-10 and CIFAR-100-20 datasets are shown in Appendix F. Given the high correlation of our objective and human labeling, we aggregate top-$n$ of labelings w.r.t. generalization error and compute the corresponding point on the plot. Thus, the leftmost strategy (at $x = 1$) corresponds to the one labeling that has the lowest generalization error, while the rightmost data point (at $x = 100$) corresponds to aggregation over all generated labelings, *i.e.*, the strategy we adopt in all experiments. By aggregation over multiple labelings the algorithm produces more stable predictions. Expectedly, given the high correlation between HUME's objective and human labeling, we can achieve even better performance by aggregating only top-$n$ predictions compared to our current strategy that considers all tasks; however, we do not optimize for this in our experiments. Finally, it can be seen that utilizing larger pretrained models eliminates the need of aggregation procedure, since they lead to stronger and robust performance.

**Reliable samples for semi-supervised learning (SSL).** We next aim to answer whether HUME can be used to reliably generate labeled examples for SSL methods. By generating a few reliable samples per class (pseudo-labels), an unsupervised learning problem can be transformed into an SSL problem [14]. Using these reliable samples as initial labels, any SSL method can be applied. HUME can produce such reliable samples in a simple way. Specifically, we say that a sample is reliable if *(i)* the majority of the found labelings assigns it to the same class, and *(ii)* the majority of the sample neighbors have the same label. We provide the detailed description of the algorithm in Appendix G. We evaluate the accuracy of the generated reliable samples on the CIFAR-10 dataset and show results in Figure 3 (b). In the standard SSL evaluation setting that uses 4 labeled examples per class [39, 42, 43], HUME produces samples with perfect accuracy. Moreover, even with 15 labeled examples per class, reliable samples generated by HUME still have perfect accuracy. Remarkably, in other frequently evaluated SSL settings on the CIFAR-10 dataset [39, 42, 43] with 25 and 400 samples per class, accuracy of reliable samples produced by HUME is near perfect (99.6% and 99.7% respectively). Results on the STL-10 and CIFAR-100-20 datasets and additional statistics of the reliable samples are provided in the Appendix E.

We next utilize the recent state-of-the-art SSL method FreeMatch [39] to compare the results of running FreeMatch with reliable samples produced by HUME to running FreeMatch with ground truth labeling. The results in Table 4 show that in the extremely low data regime FreeMatch with reliable samples produced by HUME outperforms FreeMatch with ground truth labels. Indeed, FreeMatch with ground truth labels utilizes samples which are sampled uniformly at random, while

HUME's reliable samples by definition are such samples whose most neighbours belong to the same class predicted by HUME. Consequently, FreeMatch which is based on adaptive thresholding for pseudo-labeling, benefits from utilizing labeled samples for which it can confidently set the threshold, especially in a low data regime. For instance, FreeMatch with reliable samples achieves $10\%$ relative improvement in accuracy over FreeMatch with ground truth labeling on the CIFAR-10 dataset with one sample per class. Comparing FreeMatch with $4$ reliable samples produced by HUME with an original HUME method, we observe improvement of $8\%$ on the CIFAR-10 dataset, demonstrating that HUME's results can be further improved by applying SSL with HUME's reliable samples. Overall, our results strongly demonstrate that HUME is highly compatible with SSL methods and can be used to produce reliable labeling for SSL methods.

Table 4: Comparison of accuracy of FreeMatch trained using reliable samples produced by HUME and FreeMatch trained using ground truth labeling on the STL-10 and on the CIFAR-10 dataset. We apply FreeMatch using 4 and 100 samples per class on the STL-10 dataset and 1, 4, 25 and 400 samples per class on the CIFAR-10 dataset. Each experiment is run 3 times and the results are averaged.

| | STL-10 | | CIFAR-10 | | | |
|---|---|---|---|---|---|---|
| Method | 4 | 100 | 1 | 4 | 25 | 400 |
| FreeMatch w/ reliable samples | $81.3 \pm 4.3$ | $91.3 \pm 0.3$ | $91.2 \pm 5.3$ | $95.1 \pm 0.1$ | $93.3 \pm 2.5$ | $94.3 \pm 0.4$ |
| FreeMatch w/ ground truth labels | $77.8 \pm 1.1$ | $94.0 \pm 0.1$ | $83.3 \pm 9.6$ | $94.8 \pm 0.3$ | $95.1 \pm 0.0$ | $95.7 \pm 0.1$ |

# 4 Related work

**Self-supervised learning.** Self-supervised methods [44, 45, 46, 47, 48, 49, 5, 19] aim to define a pretext task which leads to learning representations that are useful for downstream tasks. Recently, contrastive approaches [1, 2, 3, 4, 6, 20] have seen a significant interest in the community. These approaches learn representations by contrasting positive pairs against negative pairs. Another line of work relies on incorporating beneficial inductive biases such as image rotation [49], solving Jigsaw puzzles [46] or by introducing sequential information that comes from video [45]. Regardless of a learning approach, a linear probe, *i.e.*, training a linear classifier on top of the frozen representations using the groundtruth labeling, is a frequently used evaluation protocol for self-supervised methods. In our work, we turn this evaluation protocol into an optimization objective with the goal to recover the human labeling in a completely unsupervised manner. In Appendix C, we show that stronger representations lead to better unsupervised performance mirroring the linear probe evaluation. Given that HUME framework is model-agnostic, it can constantly deliver better unsupervised performance by employing continuous advancements of self-supervised approaches. Furthermore, HUME can be used to evaluate performance of self-supervised methods in an unsupervised manner.

**Transfer learning.** Transfer learning is a machine learning paradigm which utilizes large scale pretraining of deep neural networks to transfer knowledge to low-resource downstream problems [50, 7]. This paradigm has been successfully applied in a wide range of applications including but not limited to few-shot learning [10, 9], domain adaptation [51, 52] and domain generalization [53, 54]. Recently, foundation models [55, 56, 11, 57, 12] trained on a vast amount of data achieved breakthroughs in different fields. The well-established pipeline of transfer learning is to fine-tune weights of a linear classifier on top of the frozen representations in a supervised manner [50]. In our framework, we leverage strong linear transferability of these representations to act as a regularizer in guiding the search process of human labeled tasks. Thus, it can be also seen as performing an unsupervised transfer learning procedure. While language-image foundation models such as CLIP [6] require human instruction set to solve a new task, HUME provides a solution to bypass this requirement.

**Clustering.** Clustering is a long studied machine learning problem [58]. Recently, deep clustering methods [59, 13, 60] have shown benefits over traditional approaches. The typical approach to clustering problem is to encourage samples to have the same class assignment as its neighbours in a representation space [13, 60, 61]. Other recent methods rely on self-labeling, *i.e.*, gradually fitting the neural network to its own most confident predictions [14, 16, 62, 63]. Alternative approaches [64, 59] train an embedding space and class prototypes to further assign samples to the closest prototype in the embedding space. In contrast, our framework redefines the way to approach a clustering problem.

Without explicitly relying on notions of semantic similarity, we seek to find the most generalizable labeling regardless of the representation space in the space of all possible labelings.

**Generalization.** One of the conventional protocols to assess an ability of the model to generalize is to employ held-out data after model training. In addition, different measures of generalization [65, 66, 67, 68, 69, 70] have been developed to study and evaluate model generalization from different perspectives. Although, traditionally a generalization error is used as a metric, recent work [17] employs test-time agreement of two deep neural networks to find labelings on which the corresponding neural network can generalize well. Often, these labelings reflect inductive biases of the learning algorithm and can be used to analyze deep neural networks from a data-driven perspective.

**Meta-optimization.** The proposed optimization procedure requires optimization through the inner optimization process. This type of optimization problems also naturally arises in a gradient-based meta-optimization [27, 71, 72]. Other works [23, 25, 24] tackle the similar problem and study how to embed convex differentiable optimization problems as layers in deep neural networks. Although meta-optimization methods are, in general, computationally heavy and memory intensive, the proposed optimization procedure only requires propagating gradients through a single linear layer. This makes HUME a simple and efficient method to find human labelings. In addition, the inner optimization problem (Eq. 5) is convex allowing for the utilization of the specialized methods such as L-BFGS [73, 74], making our framework easily extendable to further boost the efficiency.

**Semi-supervised learning.** Semi-supervised learning (SSL) assumes that along with an abundance of unlabeled data, a small amount of labeled examples is given for each class. Consequently, SSL methods [39, 45, 37, 75, 76, 38] effectively employ available labeled data to increase the model's generalization performance on entire dataset. As shown in Figure 3 (b) and in Table 4, HUME can be used to construct reliable samples and convert the initial unsupervised problem to the SSL problem. Thus, the proposed framework can also be used to eliminate the need of even a few costly human annotations, bridging the gap between supervised and semi-supervised learning.

## 5    Limitations

The instantiation of HUME employs advances from different fields of study to solve the proposed optimization problem and consequently, it inherits some limitations which we discuss below.

**Meta-optimization.** For simplicity, HUME leverages MAML [27] to solve the proposed optimization problem (Eq. 7). However, its non-convex nature prevents convergence to the labelings which attain global optima. Though, it can be seen that the aforementioned objective is a special case of well-studied bilevel optimization problems with a convex inner part. Thus, HUME can greatly benefit from utilization of specialized optimization methods [77, 78, 79, 80] to further boost the performance.

**Number of classes and empirical label distribution.** HUME assumes the number of classes and empirical label distribution is known a priori – a common assumption in existing unsupervised learning approaches [13, 14, 59]. Although, in this work, we also follow these assumptions, the proposed framework is compatible with methods which can estimate the required quantities [81, 82].

## 6    Conclusion

We introduced HUME, a simple framework for discovering human labelings that sheds a new light on solving the unsupervised learning problem. HUME is based on the observation that a linear model can separate classes defined by human labelings regardless of the representation space used to encode the data. We utilize this observation to search for linearly generalizable data labeling in representation spaces of two pretrained models. Our approach improves considerably over existing unsupervised baselines on all of the considered benchmarks. Additionally, it shows superior performance to a supervised baseline on the STL-10 dataset and competitive on the CIFAR-10 dataset. HUME could also be used to generate labelings for semi-supervised methods and to evaluate quality of self-supervised representations. Our model-agnostic framework will greatly benefit from new more powerful self-supervised and large pretrained models that will be developed in the future.

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

# A Implementation details

We utilize ResNet-18 backbone [1] as $\phi_1(\cdot)$ pretrained on the train split of the corresponding dataset with MOCOv2 [2]. Features are obtained after *avgpool* with dimension equal to 512. For the ImageNet-1k dataset, we utilize ResNet-50 backbone as $\phi_1(\cdot)$ pretrained with MOCOv2. Here, features are obtained after *avgpool* with dimension equal to 2048. We use the same backbone and pretraining strategy for baselines as well. To enforce orthogonality constraint on the weights of the task encoder we apply Pytorch parametrizations [3]. When precomputing representations we employ standard data preprocessing pipeline of the corresponding model and do not utilize any augmentations during HUME's training. In all experiments, we use the following hyperapameters: number of iterations $T = 1000$, Adam optimizer [4] with step size $\alpha = 0.001$ and temperature of the sparsemax activation function $\gamma = 0.1$. We anneal temperature and step size by 10 after 100 and 200 iterations. We set regularization parameter $\eta$ to value 10 in all experiments and we show ablation for this hyperparameter in Appendix B. To solve inner optimization problem we run gradient descent for 300 iterations with step size equal to 0.001. At each iteration we sample without replacement 10000 examples from the dataset to construct subset $(X_{tr}, X_{te}), |X_{tr}| = 9000, |X_{te}| = 1000$. Since STL-10 dataset has less overall number of samples, we use 5000 ($|X_{tr}| = 4500, |X_{te}| = 500$) in the inductive setting and 8000 ($|X_{tr}| = 7200, |X_{te}| = 800$) in the transductive setting. For the ImageNet-1000 dataset we use inner and outer step size equal to 0.1, number of inner steps equal to 100, sample 20000 examples with $|X_{tr}| = 14000, |X_{te}| = 6000$ on each iteration. We do not anneal temperature and step size for the ImageNet-1000 dataset and other hyperparameters remain the same. To reduce the gradient variance, we average the final optimization objective (Eq. 7) over 20 random subsets on each iteration. To stabilize training in early iterations, we clip gradient norm to 1 before updating task encoder's parameters. We use $N_{neigh} = 500$ in Algorithm G1 to construct reliable samples for semi-supervised learning.

# B Robustness to a regularization parameter

HUME incorporates entropy regularization of the empirical label distribution in the final optimization objective (Eq. 7) to avoid trivial solutions, *i.e.*, assigning all samples to a single class. To investigate the effect of the corresponding hyperparameter $\eta$, we run HUME from 100 random initializations $W_1$ for each $\eta \in \{0, 1, 2, 5, 10\}$ on the CIFAR-10 dataset. Figure B1 shows results with different values of hyperaparameter $\eta$. The results show that $\eta$ is indeed a necessary component of HUME objective, *i.e.*, setting $\eta = 0$ leads to degenerate labelings since assigning all samples to a single class is trivially invariant to any pair of representation spaces. Furthermore, the results show that HUME is robust to different positive values of the parameter $\eta$. We set $\eta = 10$ in all experiments.

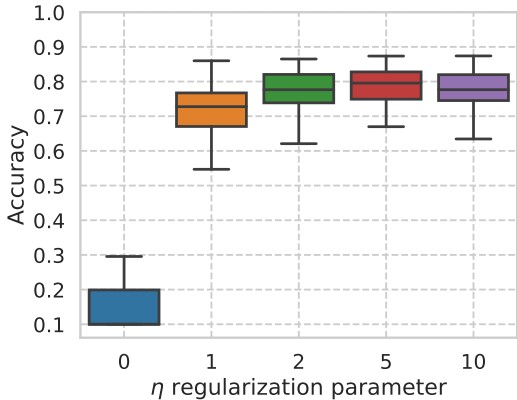

Figure B1: Performance of HUME on the CIFAR-10 dataset with different values of the entropy regularization. We use MOCOv2 self-supervised representations pretrained on the CIFAR-10 dataset and BiT large pretrained model.

## C  Ablation study on different self-supervised methods

In all experiments, we use MOCOv2 [2] self-supervised representations. Here, we evaluate HUME's performance with different self-supervised learning method. In particular, we utilize SimCLR [5] and [6] to obtain representation space $\phi_1(\cdot)$. Table C1 shows the results in the inductive setting using DINO large pretrained model as the second representation space. The results show that HUME instantiated with MOCO consistently outperforms HUME instantiated with SimCLR on all of the datasets. This is expected result since MOCO representations are stronger also when assessed by a supervised linear probe. Alternatively, utilizing BYOL shows consistent improvements over MOCO representations. These results demonstrate that HUME can improve by employing stronger self-supervised representations. Interestingly, even with SimCLR representations HUME still outperforms unsupervised baselines pretrained with MOCOv2 in Table 2.

Table C1: Comparison of different self-supervised representations. We use DINOv2 large pretrained model. Stronger self-supervised representations lead to better performance.

| Method | STL-10 ACC | STL-10 ARI | CIFAR-10 ACC | CIFAR-10 ARI | CIFAR-100-20 ACC | CIFAR-100-20 ARI |
|---|---|---|---|---|---|---|
| **SimCLR Linear** | 85.3 | 71.2 | 87.1 | 74.4 | 70.4 | 49.7 |
| **MOCO Linear** | 88.9 | 77.7 | 89.5 | 79.0 | 72.5 | 52.6 |
| **BYOL Linear** | 92.1 | 83.6 | 90.7 | 81.0 | 77.2 | 59.3 |
| **HUME SimCLR** | 86.9 | 74.3 | 85.2 | 71.8 | 51.8 | 33.9 |
| **HUME MOCO** | 90.8 | 81.2 | 88.4 | 77.6 | 55.5 | 37.7 |
| **HUME BYOL** | 91.5 | 82.7 | 89.8 | 79.7 | 56.0 | 40.3 |

## D  Correlation plots on the STL-10 and CIFAR-100-20 datasets.

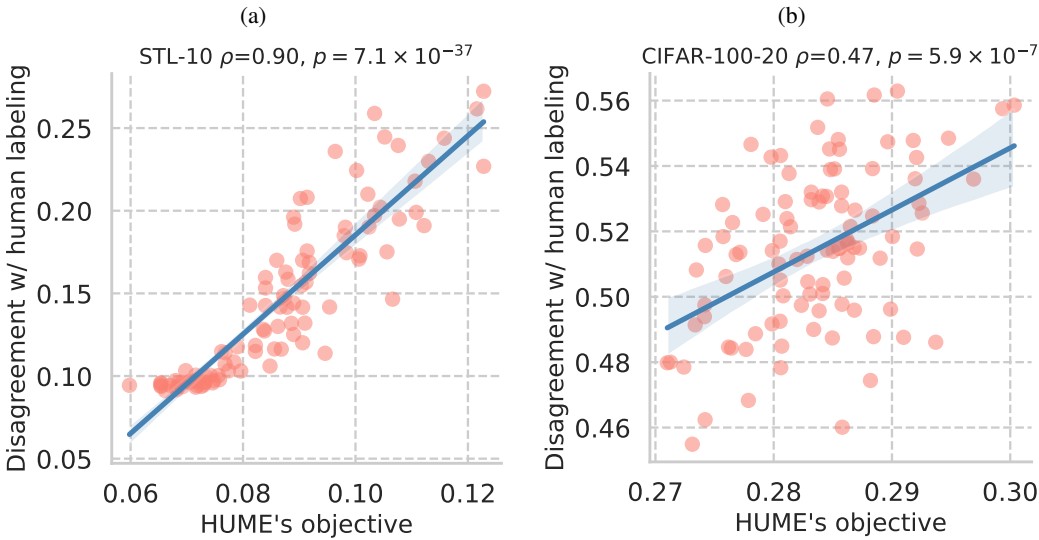

Figure D1: Correlation plot between distance to ground truth human labeling and HUME's objective on the **(a)** STL-10 dataset and **(b)** CIFAR-100-20 dataset. HUME generates different labelings of the data to discover underlying human labeling. For each labeling (data point on the plot), HUME evaluates generalization error of linear classifiers in different representation spaces as its objective function. The value of $\rho$ corresponds to Pearson correlation coefficient and $p$ is the p-value of the corresponding two-sided test. HUME 's objective is well correlated with a distance to human labeling. In particular, tasks with lower HUME's objective tend to better match ground truth labeling, *i.e.*, $\rho = 0.9, p = 7.1 \times 10^{-37}$ on the STL10 dataset and $\rho = 0.47, p = 5.9 \times 10^{-7}$ on the CIFAR-100-20 dataset.

The HUME's objective is to search over the labelings of a dataset by minimizing a generalization error. To show the correlation between HUME's objective and the ground truth labeling, we plot correlation between generalization error of the labeling measured by cross-validation accuracy with respect to the found labeling and accuracy of the found labeling with respect to ground truth labeling. In addition to the results on the CIFAR-10 dataset presented in the main paper, Figure D1 shows the correlation plots on the STL-10 and CIFAR-100-20 datasets. The results demonstrate that HUME achieves the lowest generalization error for tasks that almost perfectly correspond to the ground truth labeling on the STL-10 dataset, allowing HUME to recover human labeling without external supervision. On the CIFAR-100-20 dataset even the supervised linear model on top of MOCOv2 self-supervised representations does not attain low generalization error (72.5% accuracy in Table 1). Consequently, HUME's performance also reduces, thus this additionally suggests that employing stronger representations will lead to better performance of HUME as also shown in Appendix C. Nevertheless, Figure D1b shows fairly-positive correlation ($\rho = 0.47, p = 5.9 \times 10^{-7}$) between distance to ground truth human labeling and HUME's objective, thus confirming the applicability of HUME to more challenging setups when one of the representation spaces might be insufficiently strong.

# E Quality of the reliable samples produced by HUME

HUME can be used to produce reliable samples which can be further utilized with any semi-supervised learning (SSL) method. To measure the quality of reliable samples, we use two different statistics of the produced reliable samples: per class balance and per class accuracy. Per class balance measures number of samples for each ground truth class, *i.e.*, $\sum_{j \in R}[y_j = k]$, where $R$ is the set of indices of produced reliable samples, $y_j$ is the ground truth label of sample $j$, $k$ represents one of the ground truth classes number and $[\cdot]$ corresponds to Iverson bracket. Per class accuracy measures the average per class accuracy of the corresponding set of the reliable samples with respect to ground truth labeling. We follow standard protocol for the evaluation of SSL learning methods [7] and consider 4, 100 samples per class on the STL-10 dataset and 1, 4, 25, 400 samples per class on the CIFAR-10 dataset. We provide results averaged across all classes in Table E1 and the corresponding standard deviations across classes in Table E2. In addition to the provided statistics, Figure E1 presents accuracies of the reliable samples on the STL-10 and CIFAR-100-20 datasets for wider range of number of reliable samples per class. Overall, the results show that on the STL-10 and CIFAR-10 datasets HUME shows almost perfect balance and mean per class accuracy, *i.e.*, up to 100 samples per class on STL-10 and up to 400 samples per class on CIFAR-10, thus demonstrating that HUME can produce reliable pseudo-labels for SSL methods.

Table E1: Mean per class balance and mean per class accuracy for the reliable samples produced by HUME on the STL-10, CIFAR-10 and CIFAR-100 datasets. Mean is computed over the number of classes in the corresponding dataset.

| Quantity | STL-10 | | CIFAR-10 | | | | CIFAR-100-20 | | | |
|---|---|---|---|---|---|---|---|---|---|---|
| | 4 | 100 | 1 | 4 | 25 | 400 | 1 | 10 | 50 | 100 |
| **Mean Per Class Balance** | 4.0 | 100.0 | 1.0 | 4.0 | 25.0 | 400.0 | 0.6 | 6.3 | 26.3 | 52.6 |
| **Mean Per Class Accuracy** | 100.0 | 99.6 | 100.0 | 100.0 | 99.6 | 99.7 | 72.7 | 62.3 | 51.1 | 48.9 |

Table E2: Standard deviations of per class balance and per class accuracy for the reliable samples produced by HUME on the STL-10, CIFAR-10 and CIFAR-100 datasets. Standard deviations are computed over the number of classes in the corresponding dataset.

| Quantity | STL-10 | | CIFAR-10 | | | | CIFAR-100-20 | | | |
|---|---|---|---|---|---|---|---|---|---|---|
| | 4 | 100 | 1 | 4 | 25 | 400 | 1 | 10 | 50 | 100 |
| **Mean Per Class Balance** | 0.0 | 1.4 | 0.0 | 0.0 | 0.5 | 1.3 | 0.6 | 5.2 | 23.6 | 45.6 |
| **Mean Per Class Accuracy** | 0.0 | 0.9 | 0.0 | 0.0 | 1.2 | 0.4 | 44.1 | 41.9 | 46.5 | 47.2 |

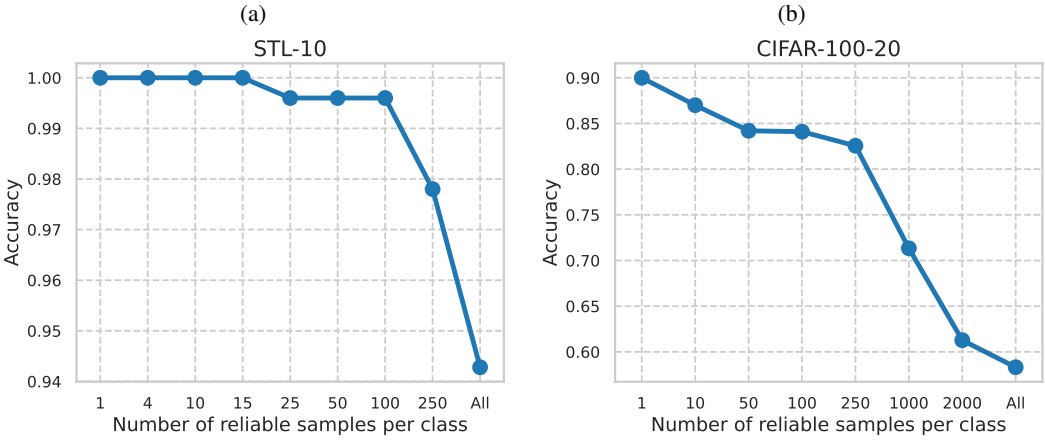

Figure E1: Accuracy of the reliable samples on the **(a)** STL-10 dataset and **(b)** CIFAR-100-20 dataset.

# F  Ablation study on different aggregation strategies on the STL-10 and CIFAR-100-20 datasets

We additionally study the effect of the proposed aggregation strategy on the STL-10 and CIFAR-100-20 datasets in an inductive setting. We employ MOCOv2 self-supervised representations as representation space $\phi_1(\cdot)$ and show the results for different large pretrained models as representation space $\phi_2(\cdot)$. Figure F1 shows the results for the STL-10 and CIFAR-100-20 datasets, respectively. We observe the similar behaviour to the results obtained in the main paper on the CIFAR-10 dataset. Namely, the proposed aggregation strategy stabilizes the results and provides robust predictions. It is worth noting that even using top-5 labelings in the majority vote is enough to produce stable results. For weaker models such as BiT, aggregation strategy has more effect on the performance and the optimal strategy is to aggregate around top-10 tasks. This is expected given the high correlation between HUME's objective and accuracy on ground truth labels since this strategy gives robust performance and tasks are closer to human labeled tasks. Larger models such as DINO show high robustness to the aggregation strategy. It is important to emphasize that in experiments we always report average across all tasks and do not optimize for different aggregation strategies.

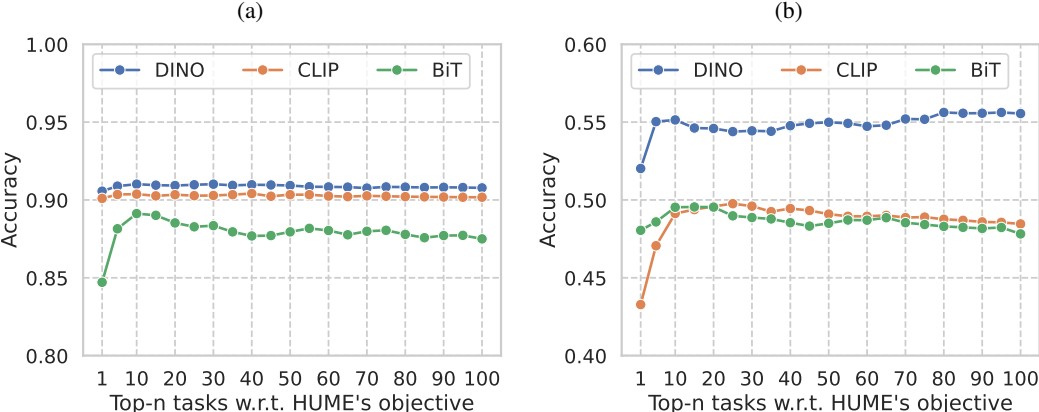

Figure F1: Different aggregation strategies on the **(a)** STL-10 dataset and **(b)** CIFAR-100-20 dataset. We use MOCOv2 self-supervised representations pretrained on the corresponding dataset and each line on the plot corresponds to the type of the large pretrained model.

# G    Algorithm for constructing reliable samples

We showed that HUME can be utilized to construct a set of reliable labeled examples to transform an initial unsupervised learning problem to a semi-supervised problem. It is worth noting that standard semi-supervised setting requires a balanced labeled set, *i.e.*, equal number of labeled samples for each class. For simplicity, we adapt the approach presented in SPICE [8] to produce a balanced set of reliable samples. Namely, we sort all samples per class by *(i)* number of labelings in the majority vote which predict the same class, and *(ii)* number of neighbours of the sample which have the same class. Thus, we consider a sample reliable if both quantities are high. Finally, given the sorted order we take the required number of samples to stand as a set of reliable samples. The proposed algorithm is outlined in Algorithm G1.

---

**Algorithm G1** Reliable samples construction

---

**Input:** Dataset $\mathcal{D}$, number of classes $K$, number of samples per class $N_k$, number of neighbours $N_{neigh}$, self-supervised representation space $\phi_1(\cdot)$, trained labelings $\tau_1, \ldots, \tau_m$

1: Compute majority vote: $\tau_{\text{MAJ}}(x) = \arg \max\limits_{k=1,\ldots,K} \sum_{i=1}^{m} \mathbb{1}[\tau_i(x) = k]$

2: Count number of agreed labelings:
   $\mathcal{A}^\tau(x) = \sum_{i=1}^{m} \mathbb{1}[\tau_i(x) = \tau_{\text{MAJ}}(x)]$

3: Find nearest neighbours in representation space $\phi_1$:
   $\mathcal{N}(x) \leftarrow N_{neigh}$ nearest neighbours for sample $x \in \mathcal{D}$

4: Count number of agreed nearest neighbours:
   $\mathcal{A}^{\text{nn}}(x) = \sum_{z \in \mathcal{N}(x)} \mathbb{1}[\tau_{\text{MAJ}}(z) = \tau_{\text{MAJ}}(x)]$

5: Initialize set of reliable samples: $\mathcal{R} = \emptyset$

6: **for** $k = 1$ to $K$ **do**

7:     Take per class samples: $S_k = \{x \in \mathcal{D}|\tau_{\text{MAJ}}(x) = k\}$

8:     Sort $S_k$ in descending order by lexicographic comparison of tuples $(\mathcal{A}^{\text{nn}}(x), \mathcal{A}^\tau(x))$

9:     Take top-$N_k$ samples from the sorted $\hat{S}_k$ and update set of reliable samples:
   $\mathcal{R} = \mathcal{R} \cup \text{top-}N_k(\hat{S}_k)$

10: **end for**

---

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
