# OpenReview forum: "The Pursuit of Human Labeling: A New Perspective on Unsupervised Learning"
_NeurIPS.cc/2023/Conference — NeurIPS 2023 spotlight_

### Official Review · Reviewer_BUAb · 2023-07-03

**Soundness:** 3 good
**Presentation:** 2 fair
**Contribution:** 3 good
**Rating:** 7
**Confidence:** 3

**Summary:**

The paper proposes a framework, called HUME, for providing human labels without any supervision. The method uses an assumption that classes derived from human labelling are linearly separable regardless of the representation space used to represent the dataset, ie invariant to sufficiently strong representation spaces. The authors show that this method generates labels that are correlated with human labeling (ie ground truth) of the dataset. HUME also shows good performance on STL-10 (ie better than a supervised linear classifier), and comparable performances on other datasets. Results are even more striking, for all the datasets used, when compared against other unsupervised methods. Finally, HUME provides reliable samples  on CIFAR-10.

**Strengths:**

From the related work explained in the paper, this is the first work in the literature to provide a method that does not rely on semantic knowledge of samples, eg clustering, but rather it uses a novel generalization perspective. The idea of using two pretrained models to generate human-like labels under an unsupervised manner is simple and interesting. The text is clear and overall well-written. The related work section is well expanded.

**Weaknesses:**

The paper could improve the analysis on the correlation of their labels with the ones humans provide, ie it’s not clear whether it generalizes to other datasets/settings, and this is a very important validation experiment of the proposed method since it’s part of the hypothesis of the paper (ie that human labeled tasks are linearly separable in a sufficiently strong representation space). Another analysis that could be added to the appendix is on the reliability of samples for other datasets (eg expands Fig. 3a of CIFAR-10 for other datasets). For now, it is only shown to work for CIFAR-10.

Minor typo: line 107 “we only train linear classifiers on top pretrained representations” should probably have an “of” between “top” and “pretrained” words.

**Questions:**

* What do the authors mean by “unbiased estimate” (line 90)?
* Which distance measure is used for fig. 1? Could this be clarified in the paper?
* Could the authors provide in appendix the same fig. 1 but for other datasets (eg CIFAR-100-20 and STL-10)? Since being well-correlated with humans is essential for the proposed method, it would be interesting to see if this result generalizes for other datasets as well.
* In the paper it is mentioned in both Definitions 1 and 2 the sentence “attains low test error” (lines 101 and 105), but how low is not commented throughout the text and this is left a bit uncommented… could the authors expand on that? Could the authors also provide some analysis/experiments on this front?
* Could the authors provide a better explanation for the difference between inductive and transductive procedures? Is it the case that the inductive one a model is learned during training, and then just the model is used to make predictions on unseen data, and in the transductive case the data is kept for the inference as well (eg using some form of nearest neighbor analysis)?
* In Table 1, could the authors provide an intuition behind why the results seem to improve more for the transductive than for the inductive procedure?

**Limitations:**

Yes, limitations have been well addressed.

---

> ### Author Rebuttal · Authors · 2023-08-09
>
> We thank the reviewer for acknowledging the simplicity of our methodology and the novelty of the generalization perspective for unsupervised learning. We are also glad that the reviewer appreciates the overall quality and soundness of our paper as well as thoroughness of the related work section. Below, we provide a detailed response to the reviewer’s questions and we hope that the provided clarifications will help in increasing the reviewer's confidence.
>
> >*The paper could improve the analysis on the correlation of their labels with the ones humans provide, ie it’s not clear whether it generalizes to other datasets/settings… Another analysis that could be added to the appendix is on the reliability of samples for other datasets*
>
> We thank the reviewer for suggesting additional analysis. In the response to the reviewer’s suggestion we include the requested evaluation. First, Fig. 1a and Fig. 1b in the rebuttal pdf shows the strong correlation ($\rho=0.9$) on the STL-10 dataset and fairly positive correlation ($\rho=0.47$) on the CIFAR-100-20 dataset. Secondly, Fig. 2a and Fig. 2b in the provided rebuttal pdf show that HUME can produce up to 100 almost perfectly accurate samples per class (>= 99% accuracy) on the STL-10 dataset and up to 100 reliable samples per class (>= 85% accuracy) on the CIFAR-100-20 dataset. We will supplement the future revision of the paper with the provided analysis to further improve the presentation of our work.
>
> >*What do the authors mean by “unbiased estimate” (line 90)?*
>
> The true risk of a model $f(\cdot)$ on task $\tau(\cdot)$ is $E_{x \sim p_{data}} \mathcal{L}(f(x), \tau(x))$. Since the true data distribution $p_{data}$ is not available we estimate it using samples from the given dataset $x_1, \dots, x_{N_{test}} \sim p_{data}$ using Eq. (1) which is unbiased estimate of the true risk. We will clarify this in the revised paper.
>
> >*Which distance measure is used for fig. 1? Could this be clarified in the paper?*
>
> We thank the reviewer for noticing the missing explanation of the used distance measure. Caption of Fig. 1 refers to this distance as a generalization error. Indeed, we compute it as follows: $\frac{1}{N_{test}}\sum_{i=1}^{N_{test}}[\tau_{gt}(x_i) \neq \tau(x_i)]$ where $\tau_{gt}(\cdot)$ is the ground truth labeling of the corresponding dataset and $\tau(\cdot)$ is the labeling found by HUME. We will further clarify this in the future revision of the paper.
>
> >*In the paper it is mentioned in both Definitions 1 and 2 the sentence “attains low test error” (lines 101 and 105), but how low is not commented throughout the text and this is left a bit uncommented… could the authors expand on that?*
>
> We thank the reviewer for suggesting the improvement of the presentation of our paper. By “attains low test error” we mean that the ground truth labeling achieves the lowest generalization error (Eq. 1) and the lowest value of HUME’s objective (Eq. 2) correspondingly for Definition 1 and Definition 2. Indeed, these definitions are key assumptions of HUME which are then validated using correlation plots (Fig. 1 in the main paper and Fig. 1ab in the rebuttal pdf). We will add these clarifications in the future revision of our work to further strengthen the presentation of our work.
>
> >*Could the authors provide a better explanation for the difference between inductive and transductive procedures? Could the authors provide an intuition behind why the results seem to improve more for the transductive than for the inductive procedure?*
>
> Each dataset consists of train $D_{train}$ and test $D_{test}$ splits. Inductive setting corresponds to training HUME only on $D_{train}$ and then evaluating on $D_{test}$. Since HUME is an unsupervised method and does not utilize any labels, we can also evaluate HUME in the transductive setting which corresponds to training HUME on the entire dataset $D = D_{train} \cup D_{test}$ and evaluating on $D_{test}$. Thus, transductive setting utilizes more data to train HUME expectedly resulting in better performance. We will explain the difference between inductive and transductive settings in the future revision of the paper.

---

> > ### Comment · Reviewer_BUAb · 2023-08-19
> >
> > Thanks authors for the detailed rebuttal. My concerns have been addressed, and I have updated my score accordingly.

---

> > > ### Author Response · Authors · 2023-08-19
> > >
> > > We thank the reviewer very much for the response and for raising the score. We are happy that the reviewer finds our response effective.

---

### Official Review · Reviewer_BH9W · 2023-07-03

**Soundness:** 3 good
**Presentation:** 3 good
**Contribution:** 4 excellent
**Rating:** 8
**Confidence:** 3

**Summary:**

This paper proposes a novel unsupervised learning algorithm called HUME.

HUME aims to minimize the disagreement between two linear models. The two linear models consume features from different backbones, and thus they are iteratively optimized to map difference feature spaces into the same label space.

HUME has achieved very promising results on extensive experiments conducted in this paper.



**Strengths:**

(1) This paper provides a fundamentally new view to tackle unsupervised learning by searching for consistent labelings between different representation spaces, i.e., human-labeled tasks are invariant to sufficiently strong representation spaces, it gives valuable insights to the following research works

(2) HUME outperforms existing state-of-the-art unsupervised learning methods by a large margin and can achieve comparable or better performances than supervised learning on some tasks

(3) HUME can produce reliable labels for semi-supervised learning to further boost the performance, which provides a good starting point for applications in many low-resource scenarios

**Weaknesses:**

I don't find substantial weakness from this paper.

**Questions:**

I am personally interested in applying semi-supervised learning to the reliable pseudo labels.

Could you provide the following results?

(1) Comparison of using reliable pseudo labels versus human labels

(2) Results on more datasets

This is not obligatory, and you could put the results in the appendix.

**Limitations:**

The authors have raised 2 limitations and proposed reasonable solutions for mitigation.

I don't find other limitations of this work.

---

> ### Author Rebuttal · Authors · 2023-08-09
>
> We thank the reviewer for the very positive evaluation of our work and acknowledging the fundamentally new perspective on unsupervised learning which our work proposes. We are glad that the reviewer appreciates the extensiveness of our experiments and finds our results very promising. We are also glad that the reviewer recognizes the importance of tackling the well-established unsupervised learning problem from a fresh standpoint that can facilitate future research.
>
> >*Comparison of using reliable pseudo labels versus human labels and results on more datasets*
>
> We agree with the reviewer on the importance of supplementing the evaluation with additional datasets to further strengthen the experimental part of our work. In response to the reviewer’s request we added the following additional experiments: (i) accuracy plots of reliable samples on the STL-10 and CIFAR-100-20 datasets, (ii) semi-supervised learning using reliable samples on CIFAR-10 and (iii) evaluation of HUME on large scale fine-grained ImageNet-1000k benchmark. We will add these results in the future revision of the paper.
>
> First, Fig. 2a and Fig. 2b in the provided rebuttal pdf show that HUME can produce up to 100 almost perfectly accurate samples per class (>= 99% accuracy) on the STL-10 dataset and up to 100 reliable samples per class (>= 85% accuracy) on the CIFAR-100-20 dataset. Secondly, we apply the state-of-the-art semi-supervised method FreeMatch[1] on the CIFAR-10 dataset with 1,4, 25 and 400 samples per class. We compare the performance of FreeMatch with ground truth labels to FreeMatch supervised with HUME’s reliable samples (Table 3 in the rebuttal pdf). The results show that FreeMatch with HUME’s reliable samples performs on par with FreeMatch with ground truth labels. The only exception is CIFAR-10 with 4 samples. However, HUME produces 4 perfectly accurate reliable samples per class (see Fig. 3a in the main paper) and the discrepancy in this setting can be alleviated by averaging the results across several 4 reliable samples per class as it is done in the standard SSL evaluation. Due to the time constraints during the rebuttal period and substantial time needed to run FreeMatch, we could only do a single run for each setting. We will provide the complete results with multiple runs and standard deviation in the future revision of the paper.
>
> Last but not least, we add ImageNet-1000k dataset to evaluate HUME on the challenging large scale fine-grained classification benchmark. Table 1 in the rebuttal pdf shows the significant improvement of 24% in accuracy over existing large-scale unsupervised baselines proposed in the literature, thus confirming the applicability of HUME to more challenging problems.
>
> [1] FreeMatch: Self-adaptive Thresholding for Semi-supervised Learning. ICLR 2023

---

### Official Review · Reviewer_J4o2 · 2023-07-05

**Soundness:** 3 good
**Presentation:** 3 good
**Contribution:** 3 good
**Rating:** 5
**Confidence:** 2

**Summary:**

This work proposes HUME, an unsupervised framework for inferring labels of an image dataset. Based on the assumptions of linear separability and model invariance, the authors propose an objective that utilizes two self-supervised models, one on the target dataset and another using large-scale pre-training. Specifically, a linear mapping is applied on the first model representation to generate pseudo labels, which is updated by MAML to minimize classification loss of another linear classifier on the second representation. HUME is empirically evaluated on three image datasets, outperforming deep clustering and sometimes also linear probing baselines.

**Strengths:**

- The paper is well organized, with the entire model built around two key assumptions (linear separability and model invariance). Strong justifications are provided for each component.
- Experiments are conducted on multiple dataset and models to validate the generalizability of HUME.
- Ablations studies are included on aggregation over n runs, regularization weight, self-supervised representation etc.

**Weaknesses:**

- My biggest concern is on the assumption of linear separability on pretrained representations. This is probably true for easy-to-classify datasets (e.g. CIFAR-10) but less likely for fine-grained tasks (e.g. CIFAR-100). There exists much more challenging vision datasets than the ones studied in the paper, where it may be difficult, if at all possible, to find a feature space that linearly separates all classes. I doubt HUME will scale favorably to these datasets.
	- Results in table 1 seems to confirm this suspicion, as the relative performance of HUME to linear probing drops quickly as the datasets becomes more difficult.
	- One may further argue that if the classes are already linearly separable, then only a few labeled examples per class are needed to achieve high accuracy, which should not cost substantial human effort to annotate. The most demanding tasks to annotate are often the ones for which a good representation does not exist.
- The fairness of experimental comparisons is questionable:
	- HUME makes use of large pretrained models such as DINO, unlike other unsupervised methods that are learned on target dataset only. I am not sure to what extent the improvement should be attributed to the stronger feature representation of large models. The authors can consider adding comparison to deep clustering or linear probing using large models as backbone.
	- The authors compared to a variant of SPICE without joint training ($\rm SPICE_S$). The standard SPICE outperforms HUME by jointly updating the feature model.
- If the class vocabulary is known, this also enables zero-shot classification by vision-language models (e.g. CLIP). I am curious how fully unsupervised methods like HUME compares to CLIP in assigning pseudo labels, or if the two can be combined into a stronger model (not just using CLIP ViT as a feature extractor, but facilitating its joint embedding space for image and text).
- Despite the claim that HUME can benefit semi-supervised learning, no SSL models were trained and evaluated to verify this. The accuracy of reliable samples serves as an indirect proof, but cannot replace full semi-supervised experiments, e.g., by following the joint training step in SPICE.

**Questions:**

Using 300 iterations (supp. L564) for the inner loop of MAML appears quite impractical. I would appreciate if the authors can comment on the computational cost of HUME, or correct any misunderstanding I had with the implementation.

**Limitations:**

Limitations are discussed in the paper. There is no discussion of broader societal impacts as the paper focus on foundational learning algorithms; though the authors may consider discussing the possibility of biases of large pretrained models being transferred to the pseudo labels through HUME, etc.

---

> ### Author Rebuttal · Authors · 2023-08-09
>
> We thank the reviewer for valuable feedback and for appreciating the coherence of our presentation, strong justification of each component of our framework and extensiveness of  our experiments.
>
> *The biggest concern of the reviewer is the assumption of linear separability of pretrained representations.*
>
> **Supervised linear probe** is a gold standard evaluation protocol in the self-supervised/weakly supervised representation learning (e.g., [1-3])]. This whole research area assumes that the produced representations are strong enough to be separated by a linear classifier. For example, [2] shows that supervised linear probe on top of CLIP-ViT-L-14 features outperforms the best known **supervised** model on 21/27 datasets. DINOv2 [1] provides in-depth study and shows the outstanding performance of supervised linear probe on a wide variety of tasks such as domain generalization, image and video classification, fine-grained benchmarks etc. In addition to strong empirical evidence, recent works [e.g., 4-6] study the linear separability of contrastive representations from the theoretical perspective. For example, [4] theoretically analyzes when contrastive representations exhibit linear separability in unsupervised domain adaptation, while [5] provides guarantees on the downstream performance of a linear model on top of pretrained representations. All these results validate the linear separability assumption of pretrained representations. Our work provides a new view on tackling unsupervised learning from this perspective and shows viability of the proposed concepts in practice. We will add a detailed exposition of this assumption in the revised paper. We hope that these clarifications help the reviewer to reconsider the overall evaluation of our work.
>
> [1] DINOv2: Learning Robust Visual Features without Supervision. arXiv preprint 2023.
>
> [2] Learning Transferable Visual Models From Natural Language Supervision. ICML 2021
>
> [3] A Simple Framework for Contrastive Learning of Visual Representations. ICML 2020
>
> [4] Beyond Separability: Analyzing the Linear Transferability of Contrastive Representations to Related Subpopulations. NeurIPS 2022.
>
> [5] A Theoretical Analysis of Contrastive Unsupervised Representation Learning. ICML 2019
>
> [6] Contrastive estimation reveals topic posterior information to linear models. JMLR, 2021
>
> >*I doubt HUME will scale favorably to more challenging vision datasets.*
>
> To show scalability to large datasets we now included results on the ImageNet-1k dataset. Results in Table 1 (rebuttal pdf) show the remarkable improvement of 24% in accuracy over existing large-scale unsupervised baselines from the literature, thus confirming the scalability of HUME to challenging fine-grained benchmarks.
>
> >*The authors can consider adding comparison to deep clustering or linear probing using large models as backbone.*
>
> We now include an additional baseline by clustering DINO representations. Table 2 (rebuttal pdf) shows that HUME achieves 51%, 17% and 18% improvement in accuracy over this baseline on the STL-10, CIFAR-10 and CIFAR-100-20 datasets, respectively.
>
> >*Despite the claim that HUME can benefit semi-supervised learning, no SSL models were trained and evaluated to verify this.*
>
> We thank the reviewer for suggesting this analysis. Any SSL method can be applied after HUME. We now applied the state-of-the-art SSL method FreeMatch on the CIFAR-10 with 1,4,25 and 400 samples per class. We compare the performance of FreeMatch w/ ground truth (GT) labels to FreeMatch w/ HUME’s reliable samples (Table 3 in the rebuttal pdf). Results show that FreeMatch w/ reliable samples performs on par with FreeMatch w/ GT labels except on CIFAR-10 with 4 samples.However, HUME produces 4 perfectly accurate samples per class (see Fig. 3a) and the discrepancy in this setting can be alleviated by averaging the results across several 4 reliable samples per class as it is done in the standard SSL evaluation. Due to the time constraints during the rebuttal and substantial time needed to run FreeMatch, we could only do a single run for each setting. We will provide the complete results with multiple runs in the revised paper.
>
> >*The authors compared a variant of SPICE without joint training. …*
>
> SPICE includes 3 steps: (1) self-supervised pretraining, (2) clustering, and (3) joint training with SSL method. Given reliable samples produced in step (2), any SSL method can be applied in the step (3) for both SPICE and HUME. Thus, we directly compared the crucial clustering step between SPICE and HUME since this step is used to generate reliable samples and solves unsupervised learning task. Results show that HUME significantly outperforms SPICE thus consequently results in better pseudo-labels. Moreover, the additional results in the previous answer demonstrate that running SSL methods on reliable samples of HUME is comparable to running SSL on ground truth labels.
>
> >*I am curious how fully unsupervised methods like HUME compares to CLIP...*
>
> HUME can be easily applied for image-text data where $\phi_1(\cdot)$ is the text encoder and $\phi_2(\cdot)$ is the image encoder. We agree that it is an interesting research direction to combine the two but it is out of scope of this work focused on single-modality data and unsupervised learning setting compared to zero-shot in CLIP. We leave this for future work.
>
> >*Using 300 iterations for the inner loop of MAML appears quite impractical…*
>
> HUME’s inner optimization is not sensitive to the hyperparameters since HUME’s inner optimization is a convex optimization problem. For simplicity, HUME utilizes cold-start bilevel optimization (BLO) [1] but thanks to the form of HUME’s objective, any sophisticated BLO algorithm, e.g. [2], can be applied to improve the convergence rate and reduce iteration complexity of HUME.
>
> [1] On Implicit Bias in Overparameterized Bilevel Optimization. ICML 2022
>
> [2] Efficiently Escaping Saddle Points in Bilevel Optimization. arXiv 2023

---

> > ### Comment · Reviewer_J4o2 · 2023-08-18
> >
> > Thank you for their detailed response! The additional results are encouraging and addressed my primary concerns, and I have updated my rating to borderline accept. I feel that some of the rebuttal experiments (ImageNet, DINO w/ clustering, SSL) should be expanded and included in the paper to make the arguments more convincing.
> >
> > I am still a bit concerned that unsupervised learning is less critical when strong representation spaces exist ("if the classes are already linearly separable, then only a few labeled examples per class are needed to achieve high accuracy, which should not cost substantial human effort to annotate" in original review), though I see the potential of HUME as a bridge between unsupervised and semi-supervised learning in eliminating the need to label even a few examples. I would love to know the authors' view on this.
> >
> > Overall I am cautiously optimistic that this work presents sufficient contributions to be accepted to NeurIPS.

---

> > > ### Author Response · Authors · 2023-08-18
> > >
> > > > Thank you for their detailed response! The additional results are encouraging and addressed my primary concerns, and I have updated my rating to borderline accept. I feel that some of the rebuttal experiments (ImageNet, DINO w/ clustering, SSL) should be expanded and included in the paper to make the arguments more convincing.
> > >
> > > We thank the reviewer for the response and for raising the score. We are glad that our clarifications helped to increase the reviewer's confidence in our work. We will include these additional experiments in our paper and we thank the reviewer for the suggestions.
> > >
> > > Below we address the remaining concern of the reviewer. We hope that our response will further help to increase the reviewer's confidence in our work.
> > >
> > > >I am still a bit concerned that unsupervised learning is less critical when strong representation spaces exist ("if the classes are already linearly separable, then only a few labeled examples per class are needed to achieve high accuracy, which should not cost substantial human effort to annotate" in original review), though I see the potential of HUME as a bridge between unsupervised and semi-supervised learning in eliminating the need to label even a few examples. I would love to know the authors' view on this.
> > >
> > > We thank the reviewer for an additional question. In response to the reviewer’s feedback, we have conducted additional experiments that compare HUME to few-shot learning baselines. Namely, starting from the strongest DINO representation space, we employ two different few-shot baselines: (i) following [1], we compute a prototype for each class on top of DINO representations using few-labeled examples and then assign examples to the nearest prototype (we refer to it as proto baseline), and (ii)  we train a linear classifier on top of DINO representations using a few labeled examples (we refer to it as linear baseline). Remarkably, the results show that HUME consistently outperforms these two baselines in the 1-shot and 2-shot learning settings on all datasets. In the 3-shot learning setting HUME outperforms these baselines on the STL-10 dataset, matches performance on the CIFAR-10 dataset, and these baselines show better performance only on the CIFAR100-20 which in this setting requires *60 labeled examples*.
> > >
> > > In particular, in the 1-shot learning setting HUME significantly outperforms proto baseline by 39%, 24% and 44% on the STL-10, CIFAR-10 and CIFAR-100-20 datasets, respectively. Similarly, HUME outperforms linear baseline by 39%, 23% and 37% on the STL-10, CIFAR-10 and CIFAR-100-20 datasets, respectively. In the 2-shot learning setting, HUME outperforms proto baseline by  18%, 7% and 6% on the STL-10, CIFAR-10 and CIFAR-100-20 datasets, and linear baseline by 22%, 8% and 4%, respectively. It is important to emphasize that in this setting these baselines use *20 labeled examples* for STL-10 and CIFAR-10, and *40 labeled examples* for CIFAR-100-20 while HUME works *fully unsupervised*. Only, in the 3-shot learning setting we start seeing the benefits of few-shot baselines but only on the CIFAR-100-20 dataset. In particular, on the STL-10 dataset HUME still outperforms proto and linear baselines by 9% and 11% respectively, and matches performance of these baselines on the CIFAR-10 dataset, and proto and linear baselines outperform HUME by 11% and 13% on CIFAR-100-20 dataset by using 60 labeled examples. On the STL-10 dataset, we observe that HUME keeps outperforming these baselines even in a challenging 5-shot learning setting where these baselines use 50 labeled examples. We would also like to emphasize that obtaining few-labeled examples per class is in reality hard because it requires a strong prior knowledge of knowing the complete space of all possible classes in advance and then labeling a few examples per class. In many real-world tasks, this is impractical and very difficult. We thank the reviewer for this question and we will also include few-shot baselines in our paper. We believe these results further strengthen our work.
> > >
> > > Overall, our results show that although being completely unsupervised, HUME (i) outperforms *fully supervised* linear model on top of MOCO representations on the STL-10 dataset and matches the performance on the CIFAR-10 dataset (main paper results); (ii) outperforms challenging *few-shot learning baselines* ran on top of DINO representations in the 1-shot and 2-shot learning scenarios, and (iii) *semi-supervised method FreeMatch* using reliable samples from HUME matches the performance of the FreeMatch with ground truth labels.
> > >
> > > [1] Snell et al. Prototypical Networks for Few-shot Learning. NeurIPS 2017.

---

### Official Review · Reviewer_VWGE · 2023-07-06

**Soundness:** 3 good
**Presentation:** 3 good
**Contribution:** 3 good
**Rating:** 7
**Confidence:** 3

**Summary:**

The work presents a new method for inferring human labeling without supervision. The main idea of the method is that human labels for a dataset should be linearly separable for all good representations, and relatively invariant to the representation space. The method uses bi-level optimization with an inner loop that optimizes for the pseudo-labels of a training set, and an outer loop which optimizes for the test loss of a classifier trained on the pseudo-labels. HUME achieves state-of-the-art performance on standard unsupervised learning benchmarks, CIFAR and STL-10.

**Strengths:**

- The insight that human labels should be linearly separable in any sufficiently strong representation space is original and interesting. Showing that the objective function proposed in HUME is well correlated with human labeling is surprising, and would likely be useful to future work.

- HUME outperforms state-of-the-art on standard unsupervised learning benchmarks by a fair margin and impressively outperforms the supervised linear probe on STL-10.


- The paper is well organized and the writing is clear.


**Weaknesses:**

HUME uses DINO and MOCO while other methods only use MOCO. It could be that DINO is a better representation given that HUME with DINO outperforms HUME with other representations. It’s also unclear whether the performance gain is coming from ensembling multiple models or the optimization procedure of HUME. It would be useful to see an ablation where HUME is trained only on a MOCO representation for a straight-forward comparison to other unsupervised methods and to understand how much each component of HUME contributes.

As stated in the limitations, the number of classes and label distribution must be known a priori. Other works in this subfield make the same assumption so this weakness is not particular to HUME.


**Questions:**

- Can the method scale to larger datasets considering that second-order gradients need to be calculated? Does approximating the Hessian affect performance?

- MAML is known to be difficult to optimize with respect to hyper-parameters. Does this make HUME sensitive to hyper-parameters such as the number of inner-loop steps, learning rate, etc?

- In equation 3, you constrain the weights of the linear layer, $W_1$, to be orthonormal, but I didn’t see a mention of the method you’re using to do so. What is the reasoning for this design choice?


**Limitations:**

Limitations are well addressed by the authors.

---

> ### Author Rebuttal · Authors · 2023-08-09
>
> We thank the reviewer for the positive evaluation of our work and for acknowledging the originality of the proposed method as well as its significance for future research. We are glad that the reviewer finds the results impressive and our paper well written.
>
> >*HUME uses DINO and MOCO while other methods only use MOCO…*
>
> To show that the increase in performance comes from the HUME’s optimization procedure, we included  an additional baseline which, like HUME, utilizes both DINO and MOCO representation spaces at the same time (Table 2  in the rebuttal pdf) . We generate embeddings from DINO and MOCO, concatenate these embeddings and then perform clustering on top of concatenated embeddings. Results show that HUME achieves 18%, 9% and 8% improvement in accuracy over the baseline on the STL-10, CIFAR-10 and CIFAR-100-20 datasets, respectively. We will add these results in the revised paper.
>
> >*It’s also unclear whether the performance gain is coming from ensembling multiple models or the optimization procedure of HUME.*
>
> We thank the reviewer for the insightful question. The results in Fig. 3b and Fig. D1 in the Appendix D show the ablations of using different aggregation strategies (ensembling).  The leftmost strategy (at x = 1) corresponds to a single labeling (w/o ensembling), while the rightmost data point (at x = 100) corresponds to aggregation over all models. Results show that utilizing stronger pretrained models eliminates the need of ensembling many models, since they lead to more robust performance. Additionally, given the high correlation between HUME’s objective and human labeling, HUME achieves even better performance by aggregating only the top few models compared to our current strategy that aggregates all models. However, we do not optimize for this in our experiments since our goal is not to report the best results HUME can achieve but rather to demonstrate the correlation between HUME’s objective and ground truth labeling. In response to the reviewer's feedback, we will provide these clarifications in the revised paper.
>
> >*It would be useful to see an ablation where HUME is trained only on a MOCO representation …*
>
> We thank the reviewer for suggesting this evaluation; however, it is not applicable to HUME as it leads to a degenerate solution. The reason is that the key idea of HUME is to jointly use **two different** representation spaces to find ground truth data labeling over all possible labelings. HUME utilizes two representation spaces $\phi_1(\cdot)$ and $\phi_2(\cdot)$. $\phi_1(\cdot)$ is used by the task encoder $\tau(\cdot) = W_1 \frac{\phi_1(\cdot)}{\|\phi_1(\cdot)\|_2}$ to propose labelings of the data using linear layer $W_1$ at each iteration of the method. $\phi_2(\cdot)$ is used to assess the generalizability of the current proposal $\tau$ using linear layer $W_2$ in the representation space $\phi_2(\cdot)$. Since task encoder $\tau$ proposes labelings using linear layer $W_1$, **utilizing the same representation space** $\phi_2(\cdot)$, i.e., $\phi_2(\cdot) = \phi_1(\cdot)$, **will automatically lead to degenerate solutions**, i.e. $\forall\ W_1, \ W_2 = W_1$ can be easily inferred using logistic regression (inner optimization Eq. (4)) and, thus, will always be a generalizable task, which will prevent HUME from finding the ground truth labeling. We will explain this and clarify the importance of utilizing two **different** representation spaces in the revised paper.
>
> >*Can the method scale to larger datasets …*
>
> HUME is a scalable method that can be applied to the dataset of any size. The inner optimization problem in HUME is multiclass logistic regression solved by running stochastic gradient descent for a fixed number of iterations, thus depending only on the batch size and dimensionality of the representations rather than the dataset size. Additional experiments on large scale Imagenet-1000k benchmark provided in Table 1 (rebuttal pdf) further support wide applicability of HUME. It is also important to highlight that HUME’s optimization objective (Eq. 7) has a form of a stochastic bilevel optimization (SBO) problem with the convex inner part. Thus, MAML can be replaced by any sophisticated convex SBO algorithm which is a rapidly developing research area [1, 2]. We will comment on this in the revised paper.
>
> [1] Huang et al. Efficiently Escaping Saddle Points in Bilevel Optimization. arXiv 2023
>
> [2] Dagreou et al. A framework for bilevel optimization that enables stochastic and global variance reduction algorithms. NeurIPS 2022
>
> >*MAML is known to be difficult to optimize with respect to hyper-parameters. Does this make HUME sensitive to hyper-parameters …?*
>
> HUME’s inner optimization is not sensitive to the hyperparameters since HUME’s inner optimization is a convex optimization problem, i.e. multiclass logistic regression, thus it does not require extensive hyperparameter search. In our preliminary experiments on the CIFAR-10 we set a learning rate to 0.001 and number of iterations equal to 300 and fix them across all experiments.
>
> >*In equation 3, you constrain the weights of the linear layer, $W_1$, to be orthonormal, but I didn’t see a mention of the method you’re using to do so. What is the reasoning for this design choice?*
>
> Orthogonal weight matrix $W_1$ is parametrized using a product of elementary Householder reflectors, i.e., $W_1 = H_1 H_2 \dots H_m$, where each $H_i = I - \tau v_i v_i^T$. We utilize the standard PyTorch parametrizations module making the implementation straightforward in practice. We will add these details in the revised paper. This design choice  may be viewed as learning prototypes for each class which is an attractive modeling choice for the representation space $\hat{\phi}_1(\cdot) = \frac{\phi_1(\cdot)}{\|\phi_1(\cdot)\|_2}$. Thus, $W_1 \hat{\phi}_1(\cdot)$ corresponds to the cosine similarities between class prototypes $W_1$ and the encoding of the sample $\hat{\phi}_1(\cdot)$ (lines 138-142).

---

> > ### Comment · Reviewer_VWGE · 2023-08-14
> > **Response to Author Rebuttal**
> >
> > I thank the authors for their response. My concerns have been addressed and all details have been clarified. The insight of using linear separability across representations to derive human-like labels is interesting and I maintain my score of accept.

---

> > > ### Author Response · Authors · 2023-08-14
> > >
> > > We thank the reviewer for the response and for keeping confidence in our work. We are glad that our clarifications effectively addressed reviewer’s concerns. We are also happy to hear that the reviewer finds the key idea of our work interesting and useful for future work in this field.

---

### Official Review · Reviewer_uZTE · 2023-07-07

**Soundness:** 3 good
**Presentation:** 4 excellent
**Contribution:** 4 excellent
**Rating:** 8
**Confidence:** 4

**Summary:**

The authors propose a clever method that exploits recent findings about the effectiveness of linear probing across diverse representation spaces in order to discover semantically-meaningful clusters (i.e. ones that correspond to human labeling) by seeing which linearly-separable clusters are preserved across multiple representation spaces.

**Strengths:**

- HUME is a very clever method that beautifully exploits recent findings about the effectiveness of linear probing across diverse representation spaces.
- The paper is well-written and easy to follow.

**Weaknesses:**

- It's unclear whether performance gains are driven by HUME or by the strength of the second representation space. Additional baseline experiments with linear probing and clustering on top of the large pre-trained model representations (DINOv2, CLIP, ViT) would help clarify this. Another nice baseline evaluation that could help would be to simply concatenate the embeddings from the two representation spaces and repeat the linear probing and clustering evaluations on that.



**Questions:**

- When producing reliable samples for SSL, what is the class balance? The usefulness is impacted significantly by the diversity of the samples HUME can produce. I see in Fig 3a) that samples are produced from each class, but is performance roughly the same across all classes?
- Are all three of the large-pretrained models that you use pre-trained in unsupervised ways (e.g. none were ever actually trained on the ImageNet classes)?
- Unrelated to the review/scoring and purely out of curiosity, why did you name your method HUME? Is it an acronym for something?

**Limitations:**

See weaknesses and questions sections.

---

> ### Author Rebuttal · Authors · 2023-08-09
>
> We thank the reviewer for the positive evaluation of our work and for acknowledging the methodological novelty of our method. We are glad that the reviewer finds HUME to be *a very clever method that beautifully exploits recent findings about the effectiveness of linear probing* as well as our paper to be  well-written and easy to follow. Below, we address the reviewer's concerns and we hope that our response helps to increase the reviewer's confidence.
>
> >*It's unclear whether performance gains are driven by HUME or by the strength of the second representation space. Additional baseline experiments with linear probing and clustering on top of the large pre-trained model representations (DINOv2, CLIP, ViT) would help clarify this. Another nice baseline evaluation that could help would be to simply concatenate the embeddings from the two representation spaces and repeat the linear probing and clustering evaluations on that.*
>
> We thank the reviewer for suggesting to incorporate additional baselines. Based on the reviewer’s suggestion, we now included two additional baselines (1) clustering on top of the DINO representation space as the strongest representation space and (2) clustering on top of concatenated embeddings from DINO and MOCO. The results in Table 2 in the rebuttal pdf show that the performance gains are driven by HUME. Compared to DINO, HUME achieves an improvement of 51%, 17% and 18% in accuracy on the STL-10, CIFAR-10 and CIFAR-100-20, respectively.  Compared to DINO+MOCO, HUME achieves 18%, 9% and 8% relative improvement in accuracy on the STL-10, CIFAR-10 and CIFAR-100-20, respectively. We will add these baselines in the future revision of the paper to further strengthen the evaluation part of our work.
>
> >*When producing reliable samples for SSL, what is the class balance? The usefulness is impacted significantly by the diversity of the samples HUME can produce. I see in Fig 3a) that samples are produced from each class, but is performance roughly the same across all classes?*
>
> We thank the reviewer for suggesting additional analysis of reliable samples produced by HUME. The detailed statistics requested by the reviewer are provided in the Table below. We follow standard protocol for the evaluation of semi-supervised learning methods [1] and consider 4, 100 samples per class on the STL-10 dataset and 1, 4, 25, 400 samples per class on the CIFAR-10 dataset. On the STL-10 and CIFAR-10 datasets HUME shows almost perfect balance and mean per class accuracy, i.e., up to 100 samples per class on STL-10 and up to 400 samples per class on CIFAR-10.
> **Per Class Balance** measures number of samples for each ground truth class, i.e., $\frac{1}{K}\sum_{k=1}^{K}\sum_{j \in R} [y_j = k]$ where $R$ is the set of indices of produced reliable samples and $y_j$ is the groundtruth label of sample $j$.
> **Per Class Accuracy** measures the average per class accuracy of the corresponding set of the reliable samples. The number in the brackets in the dataset title corresponds to the number of reliable samples per class.
> |                    | STL-10 (4)  | STL-10 (100) | CIFAR-10 (1) | CIFAR-10 (4) | CIFAR-10 (25) | CIFAR-10 (400) | CIFAR-100-20 (1) | CIFAR-100-20 (10) | CIFAR-100-20 (50) | CIFAR-100-20 (100) |
> |--------------------|--------------|---------------|--------------|--------------|---------------|----------------|------------------|-------------------|-------------------|--------------------|
> | Per Class Balance  | 4.0 +- 0.0   | 100.0 +- 1.4  | 1.0 +- 0.0   | 4.0 +- 0.0   | 25.0 +- 0.5   | 400.0 +- 1.3   | 0.6 +- 0.6       | 6.3 +- 5.2        | 26.3 +- 23.6      | 52.6 +- 45.6       |
> | Per Class Accuracy | 100.0 +- 0.0 | 99.6 +- 0.9   | 100.0 +- 0.0 | 100.0 +- 0.0 | 99.6 +- 1.2   | 99.7 +- 0.4    | 72.7 +- 44.1     | 62.3 +- 41.9      | 51.1 +- 46.5      | 48.9 +- 47.2       |
>
> We thank the reviewer for the insightful question and we will include these results in the final revision of the paper.
>
> [1] Wang et al. FreeMatch: Self-adaptive Thresholding for Semi-supervised Learning. ICLR 2023
>
> >*Are all three of the large-pretrained models that you use pre-trained in unsupervised ways (e.g. none were ever actually trained on the ImageNet classes)?*
>
> CLIP and DINO are pretrained in an unsupervised way using contrastive learning based methods. CLIP ViT-L/14 is pretrained on WebImageText and DINOv2 ViT-g/14 is pretrained on LVD-142M. The only model pretrained in the supervised manner is BiT-M-R50x1 pretrained on ImageNet-21k. However, although BiT-M was pretrained in a supervised regime it shows the worst performance of the three considered large pretrained models and DINO and CLIP both show much better transferability of the representations to downstream datasets. We will comment on this in the revised paper to improve the presentation of our work.
>
> >*Unrelated to the review/scoring and purely out of curiosity, why did you name your method HUME? Is it an acronym for something?*
>
> HUME comes both as the acronym for HUMan labEling and from David Hume’s surname. David Hume was a philosopher who proposed new views on existing things. In our work, we propose a new perspective on the well established unsupervised learning field, thus reflecting the name. In addition, Hume also was an Empiricist, meaning he believed "causes and effects are discoverable not by reason, but by experience". In our work we **empirically** observe the surprising correlation between generalization error of ground truth labelings of a data and the proposed optimization objective, thus reflecting the empirical research approach.

---

> > ### Comment · Reviewer_uZTE · 2023-08-11
> >
> > Thank you for the excellent response! My concerns have been resolved, and I have updated my score accordingly.
> > On a side note, the name is great and the acronym design, while definitely a bit of a stretch ("labEling"), is well-justified by the motivation and not even the most egregious acronym I've seen this reviewing season :-)

---

> > > ### Author Response · Authors · 2023-08-11
> > >
> > > We thank the reviewer very much for the prompt response and for raising the score. We are very happy that the reviewer finds our response effective. We are also happy to hear that the reviewer likes acronym of our method and the story behind it.

---

### Author Rebuttal · Authors · 2023-08-09

We thank the reviewers for their valuable feedback and positive evaluation of our work. We appreciate fruitful suggestions of the reviewers that helped to improve the overall presentation of our work. In the response to the reviewers’ feedback, we have conducted additional experiments shown in the attached pdf and included in the response to reviewers. The attached pdf includes:

* Correlation analysis between HUME’s objective and ground truth labeling on two additional datasets, STL-10 and CIFAR-100-20 (Figure 1). The results show strong correlation ($\rho=0.9$) on the STL-10 dataset and fairly positive correlation ($\rho$=0.5) on the CIFAR-100-20 dataset.
* Analysis of accuracy of reliable samples produced by HUME on two additional datasets, STL-10 and CIFAR-100-20 (Figure 2).  The results show that HUME can produce up to 100 almost perfectly accurate samples per class (>= 99% accuracy) on the STL-10 dataset and up to 100 reliable samples per class (>= 85% accuracy) on the CIFAR-100-20 dataset.
* Evaluation of HUME on the large-scale fine-grained ImageNet-1000k benchmark (Table 1).  The results show that HUME achieves substantial improvements over existing large-scale unsupervised baselines proposed in the literature, outperforming the best baseline by 24% in terms of accuracy.
* Additional baselines constructed by (i) running clustering on top of DINO representations, and (ii) running clustering on top of representations obtained by concatenating DINO and MOCO representations (Table 2). The results show that HUME outperforms these baselines by 18%, 9% and 8% in accuracy on the STL-10, CIFAR-10 and CIFAR-100-20 datasets, respectively.
* Evaluation of semi-supervised learning method FreeMatch on reliable samples produced by HUME (Table 3). We compare FreeMatch performance on HUME”s reliable samples with FreeMatch performance on ground truth labeling. The results show that FreeMatch with reliable samples performs comparably to FreeMatch with ground truth labeling.

We provide more details and address reviewers’ comments in the individual response to reviewers. We hope that our detailed response and additional experiments will help to increase reviewers’ confidence.

---

### Decision · Program_Chairs · 2023-09-21

**Decision:**

Accept (spotlight)

**Comment:**

The reviewers all recommend acceptance. Having looked at the paper, reviews, and rebuttal, we are accepting this paper. The authors are encouraged to improve the final paper version by following reviewer recommendations, especially those raised in J4o2's initial review.